# Estimating errors in vehicle secondary aerosol production factors due to oxidation flow reactor response time

Pauli Simonen[1], Miikka Dal Maso[1], Pinja Prauda[1], Anniina Hoilijoki[1], Anette Karppinen[1], Pekka Matilainen[2], Panu Karjalainen[1], and Jorma Keskinen[1]

[1]Aerosol Physics Laboratory, Physics Unit, Faculty of Engineering and Natural Sciences, Tampere University, Tampere, Finland
[2]Dinex Finland Oy, Vihtavuori, Finland
**Correspondence:** Pauli Simonen (pauli.simonen@tuni.fi)

**Abstract.** Oxidation flow reactors used in secondary aerosol research do not immediately respond to changes in the inlet concentration of precursor gases because of their broad transfer functions. This is an issue when measuring the vehicular secondary aerosol formation in transient driving cycles because the secondary aerosol measured at the oxidation flow reactor outlet does not correspond to the rapid changes in the exhaust flow rate. Since the secondary aerosol production factor is determined by multiplying the secondary aerosol mass with the exhaust flow rate, the misalignment between the two leads to incorrect production factors. This study evaluates the extent of the error in production factors due to oxidation flow reactor transfer functions by using synthetic and semi-synthetic exhaust emission data. It was found that the transfer function-related error could be eliminated when only the total production factor of full cycle was measured using constant volume sampling. For shorter segments within a driving cycle, a narrower transfer function led to smaller error. Even with a narrow transfer function, the oxidation flow reactor could report production factors that were more than 10 times higher than the reference production factors if the segment duration was too short.

## 1 Introduction

Aerosol particles affect human health, climate and visibility (Pöschl, 2005; Seinfeld and Pandis, 2006). Organic compounds comprise approximately 20-90% of fine aerosol mass (Kanakidou et al., 2005), and a substantial fraction of organic aerosol originates from secondary aerosol formation (Zhang et al., 2007; Hallquist et al., 2009). The secondary organic aerosol (SOA) is formed in the atmosphere via oxidation of precursor gases. Resolving the total atmospheric SOA budget and the contributions from biogenic and anthropogenic sources is challenging, but it is estimated that the majority of SOA origins from biogenic sources. (Hallquist et al., 2009)

While SOA production from biogenic sources is globally higher than that of anthropogenic sources, the organic aerosol concentrations in large cities are dominated by anthropogenic SOA. High population density combined with local precursor emission sources results in significant contribution to air pollution mortality from anthropogenic SOA. (Nault et al., 2021) Nault et al. (2021) studied the health effects of anthropogenic SOA and used a set of aromatic precursor gases as a proxy for total anthropogenic organic precursor emissions in selected cities. A mass fraction of 20-62% of these emissions originated

from gasoline and diesel exhaust and fuel evaporation, which implies that vehicles are an important source of SOA in urban

environments.

Vehicular SOA production is not currently directly regulated. Since SOA originates from gaseous organic compounds, limitations for hydrocarbon (HC) emissions indirectly limit SOA production but there is no universal constant to convert the measured HC emissions to potential SOA formation in the atmosphere. Thus, reduction of HC emission does not linearly translate to reduced SOA formation. Regulating the SOA production specifically would require measuring the SOA production

factors (i.e., amount of potential SOA from emissions per fuel consumed) with smog chambers or oxidation flow reactors (OFRs).

SOA production factors (PFs) from vehicles have been measured with smog chambers by driving a driving cycle and injecting the exhaust to the smog chamber during the cycle (e.g., Gordon et al. (2014a); Platt et al. (2013)). The chambers are typically operated in batch mode, so that the oxidation in the chamber is actuated after the driving cycle is finished. The ad-

vantage of smog chambers compared to oxidation flow reactors is that the oxidant concentrations are close to ambient levels so that the photochemistry and aerosol processes resemble tropospheric conditions better. In contrast, the oxidant concentrations in OFRs are orders of magnitudes higher, which can introduce non-tropospheric effects (Peng and Jimenez, 2020). The OFRs are operated in continuous flow mode, which enables measurement of SOA production factors with good temporal resolution. Smog chamber experiments provide only the total SOA production factor of the driving cycle, while OFR measurements can

resolve how the SOA production differs between different driving conditions within the driving cycle. However, the delay caused by the residence time of the sample in the OFR complicates the calculation of SOA production factors. In this work, we address these complications.

While it is possible to measure HC and other pollutants directly from the tailpipe with only a small delay originating from the instrument response, the response time associated with a continuous SOA measurement using an OFR is significantly longer.

Considering that potential SOA is always dependent on emitted HC to some extent, a natural first approach to address this issue can be formulated as: How accurately could we estimate the HC emission by measuring HC at (non-oxidizing) flow reactor outlet instead of measuring directly from tailpipe?

When calculating the emission rates (g s$^{-1}$) or the total emission (g) of the exhaust gases, the gas concentrations in the tailpipe need to be multiplied with the exhaust flow rate. The total emission of gas $C$ (in g) is calculated by

$$C_{emitted} = \int_{t_0}^{t_f} [C]_{true}(t) Q_{exh}(t)\, dt, \tag{1}$$

where $t_0$ and $t_f$ are the start and end times of a driving cycle or event of interest, respectively, $[C]_{true}$ is the gas concentration (g m$^{-3}$) in tailpipe, $Q_{exh}$ is the volumetric exhaust flow rate (m$^3$ s$^{-1}$) and the product of $[C]_{true}$ and $Q_{exh}$ is the emission rate (g s$^{-1}$).

If the gas concentration (e.g. $[HC]$) is measured at the OFR outlet instead of the tailpipe, the emission rate and subsequently

the total emission will be affected as illustrated in Fig. 1 (assuming that the OFR UV lamps are off so that none of the HC will be oxidized). This is because the gas concentration is modified by the OFR residence time distribution (RTD): the gas

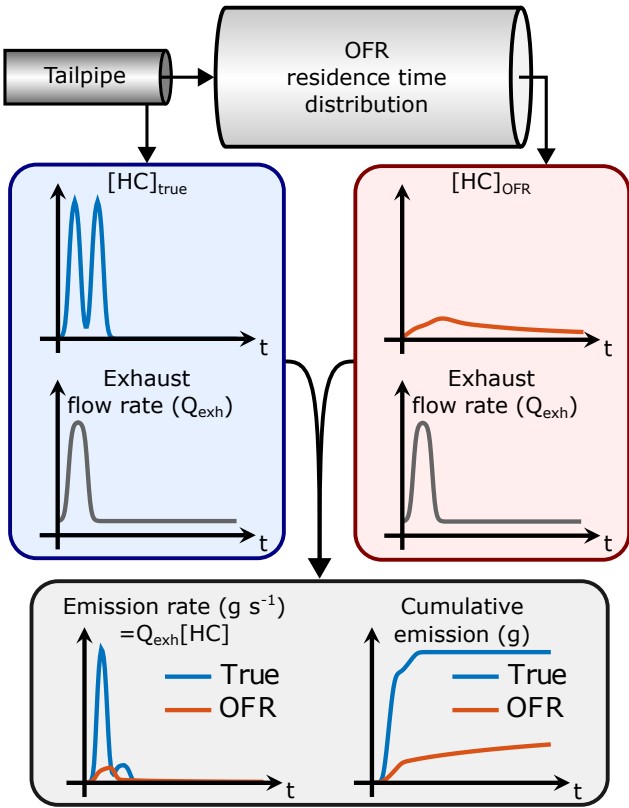

**Figure 1.** Determining HC emission by measuring HC concentration directly from tailpipe ($[HC]_{true}$) or downstream of an OFR ($[HC]_{OFR}$) and multiplying the concentrations with the engine exhaust flow rate ($Q_{exh}$). In this example, the OFR UV lamps are off so that none of the HC is oxidized. Even though the HC concentration at OFR outlet is lower because of OFR residence time distribution (RTD), the total integral is equal to that of the tailpipe HC concentration. However, the HC measured at OFR outlet will lead to underestimated HC emission because the area under OFR emission rate curve is smaller than the true area.

concentration at OFR outlet is result of convolution of the original gas concentration and the OFR transfer function, $E$. The transfer function is the RTD of a Dirac delta input impulse. (Fogler, 2006) Thus,

$$[C]_{OFR}(t) = ([C]_{true} * E)(t) = \int_0^t [C]_{true}(\gamma) E(t - \gamma) \, d\gamma, \tag{2}$$

where $[C]_{OFR}$ is the gas concentration at OFR outlet (assuming no dilution) and $[C]_{true}$ is the concentration in tailpipe. As shown in Fig. 1, multiplication of $[C]_{OFR}$ with the exhaust flow rate does not lead to the correct emission when the exhaust flow rate is not constant. Determining the correct emission would require solving for $[C]_{true}$ from the OFR measurements. Even though the transfer function $E$ can be determined, it is usually impossible to unambiguously solve for $[C]_{true}$ in Eq. 2 because this is an ill-posed inverse problem. Thus, it is not trivial to determine the SOA production factors from driving cycles

with variable driving conditions where the exhaust flow rate is not constant.

Despite the difficulties in calculating SOA PFs with OFRs, this issue is not addressed in earlier publications (Karjalainen et al., 2016; Timonen et al., 2017; Simonen et al., 2019; Pieber et al., 2018; Zhang et al., 2023). Zhao et al. (2018) recognized the problem, but there is no analysis on the magnitude of error caused by the OFR transfer function. It is necessary to estimate how large an error is caused by OFRs with different transfer functions and to determine how to best account for the transfer function when analyzing the data.

Similar issues have been studied for situations where the exhaust system, sampling lines and non-ideal instrument response cause delay and distortion to exhaust gas or particle measurements (Ganesan and Clark, 2001; Ajtay and Weilenmann, 2004; Hawley et al., 2003; Weilenmann et al., 2003; Madireddy and Clark, 2006; Geivanidis and Samaras, 2007; Franco, 2014; Giechaskiel et al., 2021). Mahadevan et al. (2016) studied the error in gaseous emission factors in test cycles due to the phenomena mentioned above. They found that the error could be as high as 51% when using non-corrected data, and 25% after applying a constant time shift to correct for the delay. The effect of delay and distortion is significantly higher for OFRs because their their dynamic response is much slower than that of gas analyzers or transportation lines.

In this study, we first present the theoretical background for calculating the SOA PF of vehicles running a transient driving cycle. Second, we study the OFR response in two real driving cycles and compare different measurement and data analysis methods. Third, we use synthetic data to further evaluate the performance of different OFRs and data analysis methods. Finally, we suggest best practices when measuring SOA PF with OFRs and provide computational tools to test the performance of any OFR for which the transfer function is known.

## 2  Theoretical background

The SOA PF defines the amount of SOA that would be formed in the atmosphere from the emitted SOA precursor gases, normalized to e.g. fuel consumed or distance travelled. Thus, the fuel-specific SOA PF $\left(\mathrm{mg\,kg_{fuel}^{-1}}\right)$ can be defined as:

$$SOA\,PF = \frac{SOA}{fuel\,consumed} = \frac{SOA}{emitted\,carbon} \cdot k' = \frac{SOA}{C_{CO_2} + C_{CO} + C_{HC} + C_{PM}} \cdot k', \tag{3}$$

where $SOA$ is the SOA formation potential $(\mathrm{mg})$, i.e., the SOA that could be formed in the atmosphere from the emitted precursor gases. The emitted carbon is the mass of carbon emitted (g), which is the sum of emitted carbon mass originating from different exhaust compounds ($CO_2$, CO, HC and particle phase carbon, $C_{PM}$) (Platt et al., 2013). The fuel consumed can be obtained from vehicle OBD data or by dividing the emitted carbon mass with the fuel carbon content denoted by $k'$ $\left(\mathrm{g\,kg^{-1}}\right)$. Since the emitted carbon is dominated by $CO_2$, it is a good approximation to neglect the other forms of carbon. For simplicity, the SOA PF in this study is defined as:

$$SOA\,PF \cong \frac{SOA}{CO_2} \cdot k, \tag{4}$$

where $CO_2$ is the emitted carbon dioxide mass and $k = k' \cdot 44/12$, i.e., the emitted $CO_2$ mass is multiplied with the ratio of carbon mass to total molecular mass in a $CO_2$ molecule.

To simplify the analysis, we assume that the SOA formation potential is single-valued and depends only on the emitted precursor gases. Furthermore, we treat the momentary SOA formation potential in the exhaust as a concentration and call this

quantity $[SOA]_{ref}$. Thus, the total SOA that is formed from the exhaust emitted in time interval $[t_0, t_f]$ is

$$SOA_{ref} = \int_{t_0}^{t_f} [SOA]_{ref}(t) \cdot Q_{exh}(t)\, dt, \tag{5}$$

where $[SOA]_{ref}$ is the momentary SOA formation potential $\left(\mathrm{mg\ m^{-3}}\right)$ in the tailpipe and $Q_{exh}$ is the exhaust gas volumetric flow rate $\left(\mathrm{m^3\ s^{-1}}\right)$. The time dependency of the SOA formation potential reflects the varying precursor gas concentrations within the driving cycle, and this will be the reference SOA to which the SOA measured downstream of the OFR will be compared.

The SOA production factor in Eq. 4 corresponds to SOA and $CO_2$ emitted within certain time interval. The momentary SOA PF is determined similarly, combining Eqs. 4 and 1:

$$SOA\ PF\ (t) = \frac{[SOA]_{ref}(t) \cdot Q_{exh}(t)}{[CO_2](t) \cdot Q_{exh}(t)} \cdot k = \frac{[SOA]_{ref}(t)}{[CO_2](t)} \cdot k. \tag{6}$$

Thus, while the SOA PF for a certain time interval depends on the exhaust flow rate, the momentary SOA PF does not. Because of this, it is not universally possible to calculate the SOA PF for a time interval based on Eq. 6 alone.

## 2.1 Determining SOA PF with an OFR

As shown in Fig. 1, the HC emission determined from HC concentration measurement at OFR outlet differs from the true HC emission. The same would be true for the part of the total HC that are the SOA precursors. As the potential SOA formation is dependent on the precursor emission, a similar error is present when measuring the SOA emission with an OFR.

Assuming that the OFR perfectly replicates the atmospheric processes that lead to SOA formation, the SOA concentration measured downstream the OFR ($[SOA]_{OFR}$) otherwise equals the SOA formation potential in the tailpipe ($[SOA]_{ref}$) but is affected by the OFR transfer function as in Eq. 2:

$$[SOA]_{OFR}(t) = ([SOA]_{ref} * E)(t) \tag{7}$$

The SOA concentration measured at the OFR outlet is delayed because of the residence time in the OFR but also distorted because of the residence time distribution as shown in Eq. 7. Ideally, the reference SOA emission could be resolved from OFR measurements by deconvolution, but the noise present in the measurement prevents a perfect deconvolution of Eq. 7. Even without the noise, it is possible that no unique solution to the inversion problem exists. Thus, it is necessary to evaluate alternative methods to estimate SOA production factor based on the distorted OFR signal.

First, to align the measured SOA concentration with the exhaust flow rate, we address the average delay caused by the OFR by shifting the OFR signal with a characteristic time constant of the OFR:

$$[C]'_{OFR}(t) = [C]_{OFR}(t + \tau_r), \tag{8}$$

where $[C]_{OFR}$ is the concentration measured downstream of the OFR and $[C]'_{OFR}$ is the delay corrected concentration, and the constant $\tau_r$ is a characteristic delay of the OFR. We chose to use the peak residence time of the OFR ($\tau_{peak}$) as $\tau_r$, i.e., the

residence time at which the transfer function reaches its maximum (see Sect. A1 and Fig. S10). An example of this correction is shown in Fig. 1, where the HC concentration at OFR outlet is delay corrected, so that the peak concentration at OFR outlet is approximately aligned with the tailpipe peak concentration. Note that this delay correction requires that the OFR start sampling zero air immediately after the cycle ends, and that the measurement downstream of OFR be continued for at least duration of $\tau_r$ for the delay corrected OFR measurement to cover the full driving cycle.

After the measured SOA concentration is synchronized with the exhaust flow rate by Eq. 8, it is possible to multiply the measured SOA concentration with the exhaust flow rate in an attempt to obtain the SOA emission:

$$SOA_{OFR} = \int_{t_0}^{t_f} [SOA]'_{OFR}(t) \cdot Q_{exh}(t)\, dt \tag{9}$$

$$= \int_{t_0}^{t_f} ([SOA]_{ref} * E)'(t) \cdot Q_{exh}(t)\, dt, \tag{10}$$

where $[SOA]'_{OFR}$ is the delay-corrected SOA concentration measured at OFR outlet that is affected by the OFR transfer function as shown in Eq. 7. Note that the exhaust sample is normally diluted before introducing it to the OFR but at this point we assume no dilution. Applying a constant dilution factor does not change the results of the analysis.

Comparison of Eqs. 10 and 5 shows that $SOA_{OFR}$ is inequal to the reference SOA emission because the delay correction does not correct for the distortion inside the OFR (Eq. 2), which leads to mismatch between exhaust flow rate and $[SOA]_{OFR}$. The only case where $SOA_{OFR}$ universally equals the reference emitted SOA is when the OFR transfer function is a Dirac delta function, i.e., when the OFR is an ideal plug-flow reactor.

## 2.2 Using constant volume sampler

The underlying issue when determining the SOA production factor is the need to multiply the SOA concentration with the exhaust flow rate. This issue is also present when determining gaseous emissions from internal combustion vehicles: because of different instrument responses and delays, there is misalignment between gas concentration values and the exhaust flow rate, causing error in the calculated emission factors (Nakamura and Adachi, 2013). The solution to this issue in regulated measurements is a constant volume sampler (CVS). Instead of trying to synchronize all gas analyzers with the exhaust flow rate data, the exhaust is diluted in a CVS with a dilution ratio (DR) that is inversely proportional to the exhaust flow rate:

$$DR_{CVS}(t) = \frac{Q_{CVS}}{Q_{exh}(t)}, \tag{11}$$

where $Q_{CVS}$ is the constant total volumetric flow rate of the CVS, which is always greater than the exhaust flow rate. The proportional dilution ratio is achieved by an arrangement where the total CVS flow is kept constant, all the exhaust is led to the CVS, and the rest of the flow required by the CVS is sampled from filtered ambient air inlet. (Nakamura and Adachi, 2013; Giechaskiel et al., 2014)

When the gas analyzers are sampling from the CVS, there is no need to multiply their concentration values with the exhaust flow rate because the dependency of emission on exhaust flow rate is already incorporated in the dilution of the CVS. Instead, the emission of a certain gas is obtained by multiplying the measured concentration with the CVS total flow.

The CVS can be utilized to OFR measurements as well to avoid the mismatch between $Q_{exh}$ and the SOA measured with an OFR, like e.g. Zhao et al. (2018), Kuittinen et al. (2021a, b) and Park et al. (2021) did. CVS should also be used in smog chamber experiments (e.g., Gordon et al. (2014a, b); Roth et al. (2020)); otherwise the injection of the exhaust into the smog chamber will not be proportional to the exhaust flow rate, i.e., the actual emission to the atmosphere.

Although the response of the OFR is much slower than that of a typical gas analyzer, it is still possible to obtain the total reference SOA emission with an OFR sampling from a CVS by multiplying the measured SOA concentration with the CVS flow rate and integrating over the full cycle:

$$\int_{t_0}^{t_f} [SOA]_{OFR}(t) \cdot Q_{CVS} \, dt = Q_{CVS} \int_{t_0}^{t_f} \left( \frac{[SOA]_{ref}}{DR_{CVS}} * E \right)(t) \, dt \tag{12}$$

$$= Q_{CVS} \int_{t_0}^{t_f} \left( \frac{[SOA]_{ref} \cdot Q_{exh}}{Q_{CVS}} * E \right)(t) \, dt = \int_{t_0}^{t_f} \left( [SOA]_{ref} \cdot Q_{exh} * E \right)(t) \, dt \tag{13}$$

$$= \int_{t_0}^{t_f} [SOA]_{ref}(t) \cdot Q_{exh}(t) \, dt \cdot \int_{t_0}^{t_f} E(t) \, dt, \quad t_0 = 0, \; t_f \to \infty \tag{14}$$

$$= \int_{t_0}^{t_f} [SOA]_{ref}(t) \cdot Q_{exh}(t) \, dt = SOA_{ref}, \quad t_0 = 0, \; t_f \to \infty. \tag{15}$$

The integral in Eq. 15 equals the integral in Eq. 5. Thus, the first integral in Eq. 12 equals the reference SOA emission. However, the separation of the convolution in Eq. 14 requires that the limits of integration be for the full defined range, i.e., $t_0 = 0$ and $t_f \to \infty$, as this is the full range of $E$ (Weisstein, 2023). In this case also the integral of $E$ is cancelled since it is unity by definition (and when dividing with emitted $CO_2$ to obtain the SOA PF, also $Q_{CVS}$ in Eq. 12 is cancelled). Thus, by using CVS it is possible to obtain the reference SOA PF for the full cycle but not for parts of it. This issue was noticed also by Zhao et al. (2018) when determining the PFs for different phases of a driving cycle. In practice, $t_f$ does not need to be infinite, but it should extend beyond the end of the driving cycle to account for the residence time in the reactor, and for this reason also the measurement of $[SOA]_{OFR}$ should be continued after the end of the driving cycle and zero air should be injected to the reactor during that time. For the driving cycles and OFRs studied here, the error in full cycle PF is less than 5% when using CVS sampling where the post-sampling duration is equal to OFR mean residence time, and the error approaches zero with longer post-sampling time (Fig. S13).

The advantage of OFRs is the continuous measurement to study the effect of different driving conditions on SOA formation. Thus, even though the CVS is a good solution for measuring the full cycle SOA PF, the applicability of OFRs for time resolved vehicular SOA studies remain unclear. The extent of the error in measured SOA emission caused by the distortion will be

studied for different scenarios in the following sections by simulating direct sampling from the tailpipe (using Eq. 9), and by simulating CVS sampling (using Eq. 12) where the integration range is significantly shorter than the full cycle length.

## 3  Methods

The analysis of the error in SOA PF arising from OFR transfer function is based on computational study where we first define a SOA reference with temporal variability and then simulate the time series of SOA concentration at OFR outlet affected by the transfer function. The SOA PF calculated based on the reference SOA is then compared to the SOA PF determined from the simulated OFR measurement. In addition to computational methods, experiments were conducted to obtain a realistic SOA reference, to characterize the OFR transfer function and to evaluate the validity of the assumptions applied in the computations.

It is currently not possible to determine a true reference for the SOA formation potential in vehicle exhaust. For the purposes of this study, it would be possible to define totally arbitrary SOA reference. However, to link the study to real exhaust emissions, we measured the time series of gaseous hydrocarbon concentration in vehicle exhaust and assumed that it represents similar temporal behaviour than the real SOA formation potential in the tailpipe. Thus, we use a simple model for the reference SOA ($[SOA]_{ref}$) in which we assume that the potential SOA is directly proportional to the measured hydrocarbon mass concentration ($[HC]$) in the tailpipe:

$$[SOA]_{ref}(t) = [HC](t) \cdot Y, \tag{16}$$

where $Y$ is the proportionality factor that includes both SOA yield and the fact that not all hydrocarbons produce SOA. Even though the proportionality factor in reality was not constant, it would mainly affect the absolute variability but not the temporal variability.

We measured HC and $CO_2$ concentrations in the exhaust of a Euro 6 gasoline vehicle running two driving cycles to obtain reference data. The HC concentration was measured with a flame ionization detector. The $CO_2$ concentration required in SOA PF calculation (Eqs. 4 and 1) was measured with a non-dispersive infrared analyzer, and the volumetric exhaust flow rate was calculated based on the intake air flow rate and fuel consumption obtained from the on-board diagnostics data. The driving cycles were cold-start (CS) New European driving cycle (NEDC) which was preceded by soaking time of 15 h and started with an engine start, and hot-start (HS) NEDC which was preceded by driving at 80 km h$^{-1}$ speed for 5 min and started with an idling engine. Proportionality factor ($Y$) of 0.15 was used in Eq. 16, resulting in realistic SOA PFs of approximately 100 and 20 mg kg$_{\text{fuel}}^{-1}$ for cold-start and hot-start cycles, respectively.

The OFR transfer function needed to be well defined to simulate its effect on the SOA production factor calculation. For this, we used a prototype of Dekati oxidation flow reactor (DOFR; Dekati Ltd) which is a commercial oxidation flow reactor with similar geometry to that of Tampere secondary aerosol reactor (TSAR; Simonen et al. (2017)). The DOFR transfer function was determined for $CO_2$ and toluene by measuring the DOFR outlet concentrations of 10 s input square pulses. To compare OFRs with different transfer functions, we also determined the transfer function of a Potential Aerosol Mass (PAM) reactor by utilizing the $CO_2$ pulse data presented by Lambe et al. (2011). Additional details on vehicle exhaust and OFR characterization measurements are provided in Appendix A.

Two different sampling options will be considered in the analysis: direct sampling from tailpipe and sampling from a CVS. The SOA concentration at OFR outlet was simulated by convolving $[SOA]_{ref}$ with OFR transfer function (Eq. 7) for cases with direct sampling. For CVS sampling, the SOA concentration at OFR outlet was simulated by similar convolution, but the varying degree of dilution of the sample entering the OFR was accounted for:

$$[SOA]_{OFR,CVS}(t) = \left( \frac{[SOA]_{ref}}{DR_{CVS}} * E \right)(t). \tag{17}$$

Both Eqs. 17 and 7 assume that the OFR otherwise reproduces perfectly the SOA reference but only with slower response. The transfer function used in these equations is the transfer function of $CO_2$. In reality, because of the following effects these assumptions do not necessarily hold:

1. Non-tropospheric gas-phase chemistry and other non-tropospheric losses (e.g. wall losses of precursor oxidation products). (Peng and Jimenez, 2020, 2017; Peng et al., 2019; Palm et al., 2016) These can vary within the driving cycle because e.g. the wall losses depend on the particle surface are concentration inside the OFR. They can also affect the OFR response in general because the sample at OFR outlet has a distribution of residence times, and some of the losses are time-dependent. Thus, the square pulse injection of SOA precursor would not produce a SOA mass concentration profile at the OFR outlet that is similar to a $CO_2$ profile produced by a square pulse injection of $CO_2$. In other words, the use of $CO_2$ transfer function would be incorrect.

2. Even though the proportionality factor in Eq. 16 was constant, the SOA yield inside the OFR can change depending on the organic aerosol mass concentration inside the OFR and on the OH exposure. As in the previous entry in this list, this may vary within the driving cycle but also in the OFR residence time dimension.

3. The amount of SOA formed depends on the amount of consumed precursor gases. Depending on the OH exposure in the OFR (assuming OH reactive precursor gas), all precursor gases do not necessarily fully oxidize. Since the OH exposure depends on the average OH concentration in the OFR and on the residence time, there will be a distribution of OH exposures at the OFR outlet because of the residence time distribution of the OFR. Thus, the shape of SOA pulse originating from an input pulse of precursor gas depends on the combination of OFR transfer function and the reaction rate constant of the precursor gas.

4. There are potential adsorption, absorption or desorption phenomena in the OFR or preceding sampling lines. Several studies have shown that gaseous organic compounds may exhibit significant delays in sampling lines or instruments when they are first adsorbed or absorbed on the sampling line or instrument wall and later desorbed (Pagonis et al., 2017; Deming et al., 2019; Liu et al., 2019; Morris et al., 2024). Similar effects may be present in an OFR as well for the precursor gases or their oxidation products, worsening the misalignment between the produced SOA and the exhaust flow rate. Morris et al. (2024) showed the effect of adsorption and absorption on the PAM reactor response time for ketones, but not for typical SOA precursors.

To keep the analysis simple, all the effects listed above are neglected, but estimation of their relevance is discussed here. We measured a rapid SOA formation pulse in DOFR by injecting a square pulse of toluene at the reactor inlet. The SOA concentration that was measured at DOFR outlet following this pulse is shown in Fig. S1a. As a comparison, a computational result, the square pulse of toluene convolved with $CO_2$ transfer function is showed in the same figure. While the computational result does not perfectly replicate the measured SOA concentration, the agreement is good and we can deduce that the effects listed above are minor for toluene SOA formation.

The average OH exposure in the toluene pulse experiment was $7.9 \cdot 10^{11}$ cm$^{-3}$ s$^{-1}$ (equivalent of approximately 6 days of OH oxidation in the atmosphere with OH concentration of $1.5 \cdot 10^6$ cm$^{-3}$). At this high OH exposure, essentially all the toluene was consumed. For slower-reacting precursor gases, such as benzene, this would not be the case as shown in Figs. S1b and S1d. However, the reaction rate constants between OH radicals and most anthropogenic SOA precursors are higher than that of toluene, indicating that the assumption that all precursor gas is consumed in the OFR is sufficient, as long as the OH exposure is approximately $7.9 \cdot 10^{11}$ cm$^{-3}$ s$^{-1}$ or higher. A more detailed discussion is presented in Supplement section S2.

To our knowledge, the delay effects caused by adsorption, absorption and desorption have not been characterized for typical SOA precursors. Even though we did not observe such delays for the toluene SOA formation in DOFR, the effect has potentially high impact on other SOA precursors, especially the less volatile ones (intermediate volatility compounds), and needs further research.

## 4 Results and discussion

To study the effect of OFR RTD on the accuracy of SOA production factor, we simulate the SOA concentration at OFR outlet for two OFRs that have distinct residence time characteristics. The PAM reactor (Lambe et al., 2011) represents an OFR with a broad transfer function, with mean residence time of 142 s and transfer function standard deviation of 113 s. A prototype version of DOFR has a faster response with mean residence time of 41 s and transfer function standard deviation of 21 s. The standard deviation of the transfer function reflects the transfer function broadness and its calculation is presented in Sect S1. All OFR data shown hereafter is delay corrected according to Eq. 8 with the peak residence time of the OFR (see Sect. A1).

### 4.1 Real driving cycles

As described in Sect. 3, in the absence of a true reference for SOA formation potential, we generated semi-synthetic data based on the HC concentration measured from a gasoline vehicle tailpipe, assuming that SOA formation potential is directly proportional to the HC concentration (Eq. 16). The $CO_2$ and SOA concentrations at OFR outlets were simulated by convolving the tailpipe concentrations with the OFR transfer functions (Eq. 2). We assumed that the reactors were sampling zero air until the cycle starts; otherwise, the exhaust from preceding driving would be present in the OFRs and affect the cycle-specific SOA PF.

Figure 2 shows the $CO_2$ concentrations and SOA concentrations, and their cumulative emissions in the hot-start NEDC. Similar graphs for the cold-start NEDC are shown in Fig. S3. The SOA concentration at the DOFR outlet follows the reference

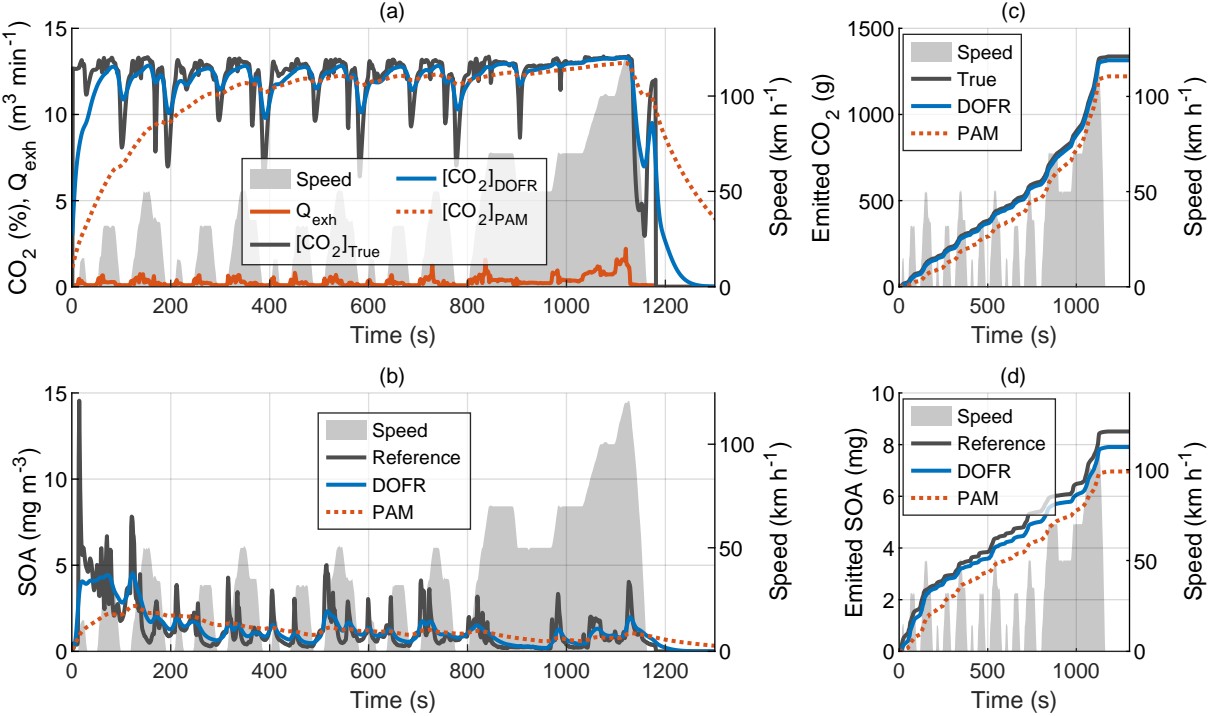

**Figure 2.** Time series of exhaust flow rate ($Q_{exh}$), tailpipe and OFR outlet concentrations of $CO_2$ (a) and SOA (b) in hot-start NEDC, and the cumulative emissions of $CO_2$ (c) and SOA (d). The OFR data is simulated based on tailpipe concentrations and OFR transfer functions, and the SOA concentration refers to HC concentration multiplied with $Y$. All OFR data is delay corrected.

SOA concentration better than PAM, which results in better agreement on the total emitted SOA at the end of the driving
cycle (Fig. 2d). However, the DOFR-based total SOA emission is still 7% lower than the true SOA emission, mainly because
the response is not fast enough to follow the reference SOA concentration during accelerations where the exhaust flow rate is
highest. PAM has the same effect, but in addition the PAM-derived SOA emission starts to deviate from the reference SOA
emission already in the beginning of the cycle because the response is too slow to catch the SOA peak in the cycle start. These
two effects result in total SOA emission that is 18% lower than the reference SOA emission.

Because the SOA PF is directly proportional to ratio of emitted SOA and emitted $CO_2$, the relative error in PF equals the
relative error in the SOA emission. However, for the OFRs both SOA emission and $CO_2$ emission (calculated from the delay
corrected $CO_2$ measured at OFR outlet) are underestimated in the driving cycles studied here, so the error in PF could be
decreased by normalizing the SOA emission to $CO_2$ emission measured at OFR outlet instead of true $CO_2$ emission. Even
though this calculation method leads to better estimation of SOA PF in the two cases studied here, it is not guaranteed that
the error in $CO_2$ measurement will always compensate for the error in SOA measurement. It is possible that in some cases
the SOA emission determined from OFR measurements is higher than the reference emission, and in such case normalizing

to OFR $CO_2$ would amplify the error. Therefore, when presenting the integrated SOA PFs (e.g. Fig. 3a), the SOA emission is normalized to true $CO_2$ emission.

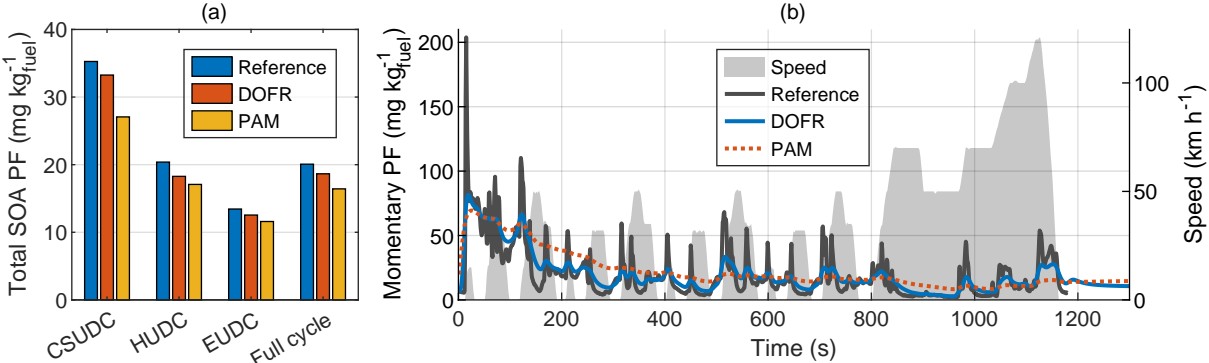

**Figure 3.** Total SOA PFs of subcycles and full driving cycle (a), and time series of reference SOA PF and SOA PFs determined from OFR outlet concentrations (b) in hot-start NEDC. The integrated SOA PF in panel (a) is calculated by normalizing the SOA emission to true $CO_2$ emission, whereas for the momentary SOA PF in panel (b), the SOA concentration is normalized to OFR $CO_2$ concentration. The PFs in both panels are calculated from semi-synthetic SOA data that is linearly proportional to the measured HC concentration in the tailpipe or the simulated HC at OFR outlet. CSUDC, HUDC and EUDC represent approximately 400 s subcycles within the full cycle.

The error in full cycle SOA PFs is relatively small for both cold- and hot-start driving cycles despite the distorting effect of OFR transfer functions. In hot-start NEDC, the error in total SOA PF is 7% for DOFR and 18% for PAM (Fig. 3), and in cold-start NEDC the corresponding errors are 4% and 7% (Fig. S4). To study the accuracy of SOA PF in smaller subcycles, we divided the NEDC into three parts according to Karjalainen et al. (2016): cold start urban driving cycle (CSUDC; 0...391 s), hot urban driving cycle (HUDC; 392...787 s) and extra-urban driving cycle (EUDC; 788...1180 s). The division is used here also for the hot-start cycle although the term CSUDC does not represent a cold start in that case. The maximum error in the subcycles was 10% for DOFR (hot-start HUDC) and 23% for PAM (hot-start CSUDC). Note that the SOA PFs for the subcycles (Fig. 3a) are not the average values of momentary PFs shown in Fig. 3b. Instead, the subcycle SOA PF is calculated by normalizing the SOA emitted during the subcycle to the emitted $CO_2$.

The continuous operation of the OFRs allows studying SOA production factors at higher time resolution than the ~400 s subcycles. Zhang et al. (2023) investigated SOA PF as a function of driving condition by using a fast-response OFR (Veh-OFR). Such analysis requires time resolution in order of seconds, and the effect of OFR transfer function on the accuracy of momentary SOA PF at such time resolution needs to be determined.

The time-resolved reference and OFR SOA PFs are shown in Fig. 3b for hot-start driving cycle and in Fig. S4b for cold-start driving cycle. The time-resolved OFR SOA PFs were calculated by normalizing the SOA concentration to $CO_2$ measured at OFR outlet to compensate for the slow response in SOA measurement. This is important especially in the beginning of the cycle, where the $CO_2$ levels in the OFRs deviate significantly from the tailpipe concentration.

Figure 3b shows that although the DOFR PF time series resembles better the reference PF time series than PAM, neither of the OFRs can follow the rapid changes of the reference SOA PF. For example, the maximum OFR PFs during the acceleration starting at 313 s are approximately 40% of the reference maximum PF. However, when integrating the SOA and $CO_2$ emissions for a longer time interval, the agreement between the reference PF and OFR PFs improves. For the full duration of the acceleration (313...343 s), the DOFR PF is 74% and PAM PF is 82% of reference PF. In general, the longer the integration time interval, the better the agreement (Fig. S11). Thus, when studying the effect of driving conditions on SOA production, it is better to divide the driving cycle in bins that represent different driving conditions instead of determining the relations based on second-by-second data.

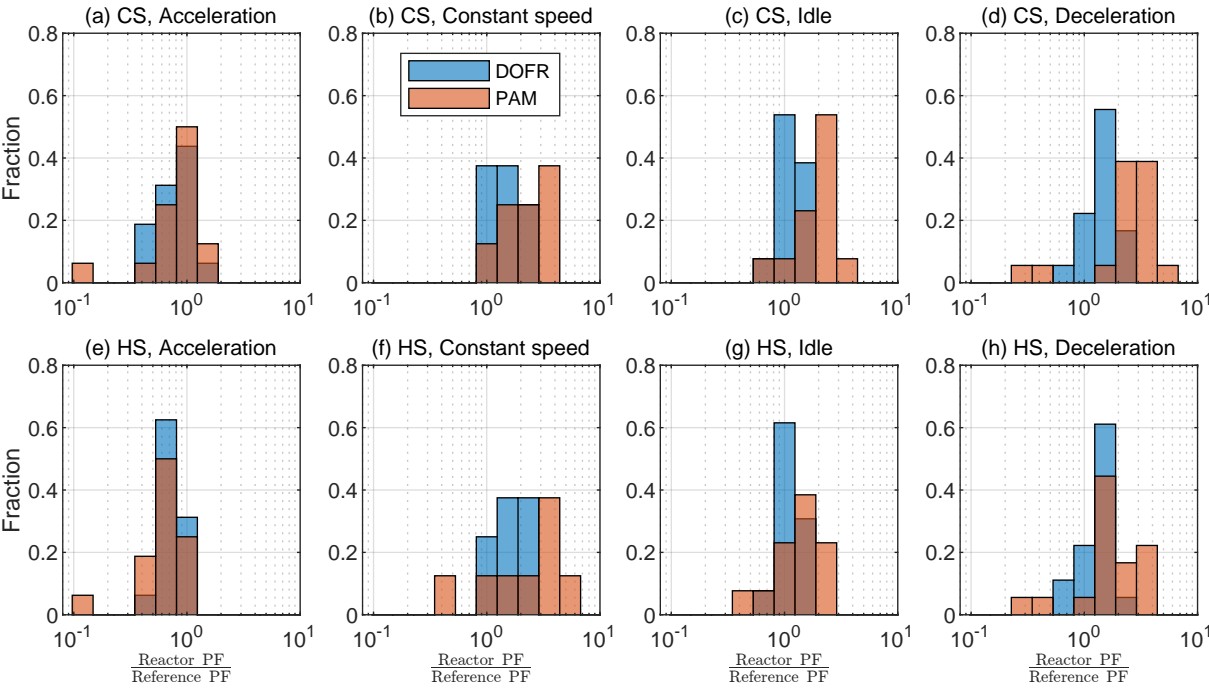

**Figure 4.** OFR SOA PF deviation from reference PF for different driving condition bins, when the SOA PF is determined by normalizing the emitted SOA to true emitted $CO_2$. The cold-start cycle is denoted as CS and hot-start as HS. Corresponding correlation plots are shown in Fig. S15. The PFs are calculated from semi-synthetic SOA data that is linearly proportional to the measured HC concentration in the tailpipe or the simulated HC at OFR outlet.

To study the accuracy of different OFRs, we divide the driving cycle in short events according to different driving conditions: accelerations, constant speed driving, idling and decelerations. The time periods are shown in Fig. S14, and the deviations from the reference SOA PF for each driving condition are shown in Fig. 4. The corresponding correlation graphs are shown in Fig. S15.

Figure 4 shows that DOFR with narrower RTD is generally better suitable for studying SOA PFs of short events than PAM. In the cold-start cycle, PAM typically overestimates the PF because the HC originating from the engine start remain in

PAM for a long time. In both cycles, the acceleration PFs are usually underestimated by both OFRs because there is typically simultaneous increase in exhaust flow rate and HC concentration, but the OFR SOA does not reach the level of reference SOA as illustrated in Fig. 1 for HC. In constant-speed driving, both OFRs overestimate SOA PF because this driving condition is usually preceded by accelerations, and HC originating from the acceleration is still present in the OFRs. For the same reason, the OFRs overestimate also the deceleration PFs.

As discussed earlier, normalizing the emitted SOA to $CO_2$ emission determined from $CO_2$ concentration measured at OFR outlet may reduce the error in SOA PF. The applicability of this method and other methods to reconcile the distortion in SOA concentration caused by the OFR transfer functions are studied in the next section.

### 4.1.1 Alternative data analysis and measurement methods

Figure 5 shows the SOA PF deviations for both reactors when using different data analysis and measurement methods. Overall,
the different methods (except for the averaging method) result in relatively small error, maximum 37 %. In all methods, the OFR data is delay corrected.

The *standard* **method** is the one used in previous sections, i.e., the SOA emission is normalized to true $CO_2$ emission. This method underestimates the SOA PF in most cases (Fig. 5). Note that normalization to true $CO_2$ emission is equivalent to normalizing to true fuel consumed or true distance travelled. When using the other data analysis methods described below
and calculating the distance based production factors, one first needs to determine the fuel-specific production factor and only then convert it to distance based by multiplying with the ratio of fuel consumed per distance travelled that is available in the on-board diagnostics data.

In the *OFR* $CO_2$ **method** the $CO_2$ concentration is measured from OFR outlet and the $CO_2$ emission is determined by multiplying the delay-corrected $CO_2$ concentration with the exhaust flow rate. In most cases, the *OFR* $CO_2$ method results in
better agreement with the reference SOA PF compared to standard method (Fig. 5), which is in agreement with the observation that both SOA and $CO_2$ emissions are underestimated with the OFR in Fig. 2.

The *convolution* **method** applies the same OFR response to the exhaust flow rate that affects the SOA and $CO_2$ concentrations that are measured at OFR outlet. The SOA and $CO_2$ emission rates are calculated by multiplying the concentrations at OFR outlet with exhaust flow rate that is convolved with OFR transfer function. This method was used by Simonen et al.
(2019) for determining SOA emission rate, but it was not normalized to $CO_2$ emission measured at the OFR outlet but to the true fuel consumption or distance travelled, which is equal to normalizing to true $CO_2$ emission. The deviation in *convolution* method is of similar magnitude to the standard method and the *OFR* $CO_2$ method (Fig. 5).

In the *CVS* **method**, the OFRs are sampling exhaust that is diluted with CVS, i.e., the dilution ratio is inversely proportional to the exhaust flow rate. The emitted SOA is calculated with Eq. 12. The emitted $CO_2$ is calculated with a similar equation,
where the $CO_2$ is measured at the OFR outlet. The *CVS* method always leads to correct SOA PF for the full cycle as discussed in Sect. 2.2. For DOFR, the *CVS* method results in least deviation in subcycles as well compared to the methods presented above. For PAM, the deviation in subcycles with this method is on average larger than the previous methods in cold-start cycle, but performs better in the hot-start cycle.

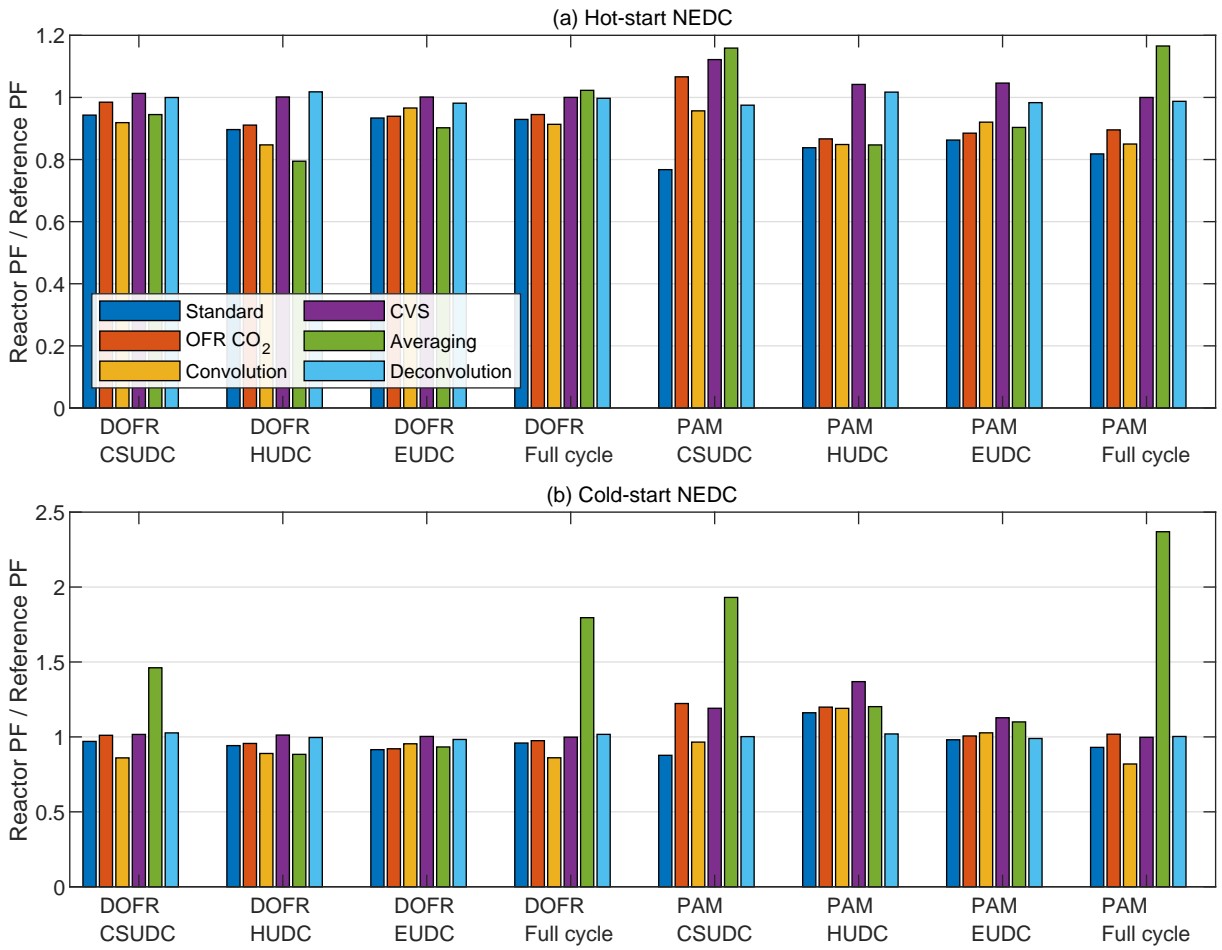

**Figure 5.** OFR SOA PF deviation from the reference PF for full driving cycles and ∼400 s subcycles when using different data analysis and measurement methods. The PFs are calculated from semi-synthetic SOA data that is linearly proportional to the measured HC concentration in the tailpipe or the simulated HC at OFR outlet.

Although the CVS sampling is favorable especially for DOFR, it has some disadvantages. CVS requires a high flow rate
of dilution air compared to partial flow diluters, and purifying such amounts of dilution air is challenging. This may lead to high background SOA formation from dilution air impurities (Zhao et al., 2018). In addition, the heat from vehicle exhaust may cause desorption of previously adsorbed HC from CVS walls (Gordon et al., 2014a). However, the SOA PFs have been measured with CVS sampling with acceptable background SOA formation (Zhao et al., 2018; Kuittinen et al., 2021a; Gordon et al., 2014a).

An inherent feature of the CVS is that the dilution ratio is inversely proportional to exhaust flow rate. As shown in Fig. 2, the HC peaks usually occur during accelerations, where the exhaust flow rate is also elevated. The same is observed for $NO_X$, so the use of CVS dilution amplifies the variations in HC and $NO_X$ concentrations compared to direct sampling from tailpipe

with constant dilution ratio. Since the OH exposure and photochemistry in OFRs is sensitive to concentrations of $NO_X$ and OH reactive gases (Peng and Jimenez, 2017), using CVS may cause too high gas concentrations during e.g. accelerations where exhaust flow rate and gas concentrations are high, and too low signal during e.g. idling where concentrations and exhaust flow rate are low.

The *averaging* **method** does not consider the exhaust flow rate, since it is calculated using Eq. 6. In the study by Zhang et al. (2023), the full cycle SOA PFs were apparently determined by calculating the average of the momentary PFs instead of calculating the ratio of emitted SOA to emitted $CO_2$, although their description of the PF calculation for full cycle is not unambiguous. Figure 5 shows that in the cold-start cycle, this *averaging* method leads to an overestimation by a factor of $\sim 2$ for DOFR and $\sim 2.5$ for PAM. Note that the PFs calculated with the *averaging* method are not compared to the average values of reference momentary PFs, but instead to the reference PF which is the emitted SOA normalized to emitted $CO_2$.

In the *deconvolution* **method**, the SOA signal simulated at OFR outlet is first deconvolved (as described by Conesa (2020); see Sect. S3) to obtain the reference SOA concentration in tailpipe, and then multiplied with true exhaust flow rate to obtain the SOA emission rate. The emitted SOA is normalized to true emitted $CO_2$. For PAM, the *deconvolution* method leads to smallest errors, whereas for DOFR the *CVS* method is as good as the *deconvolution* method. The time series of deconvolved SOA concentrations are shown in Fig. S16.

The deconvolution here represents the best possible outcome because there is no noise present in the simulated SOA concentration at OFR outlet. In real-life scenarios, there is noise originating from the instrument measuring the SOA concentration and also some variability in the OFR transfer functions due to small fluctuations in flow rate and temperature. The performance of the *deconvolution* method in such cases is beyond the scope of this study, but our tests for 10 s square pulses of SOA precursor showed that the deconvolution was able to reproduce the square pulses based on the measured SOA concentration at DOFR outlet, but not perfectly (Fig. S9).

While all calculation methods except the *averaging* method are able to report the SOA PF for full cycles and $\sim 400$ s subcycles with relatively good accuracy, Fig. 4 shows that in some cases, the deviation in short driving events can be very high when using the *standard* method. Some of the deviations in Fig. 4 could be avoided by normalizing the SOA emission to the $CO_2$ measured at OFR outlet instead of tailpipe. For example, the most severe underestimations in PAM and DOFR are observed in the beginning of the driving cycles where the OFR response to $[SOA]_{ref}$ is much slower than the change in the tailpipe $CO_2$ concentration, and in other occasions where there is drastic change in tailpipe $CO_2$ concentration. Likewise, the changes in tailpipe $CO_2$ concentration during decelerations are much faster than the characteristic residence times of the OFRs. For this reason, we investigate whether the normalization to OFR $CO_2$ or any of the other methods perform better for short events in the driving cycles. For this analysis, we divide the cycle in 14 s bins and calculate the deviation from reference PF for each bin using different methods. The 14 s bin duration was chosen because it is the median duration of different events in Fig. 4.

Figure 6 shows that different calculation methods, including the *averaging* method (but excluding the *deconvolution* method), report similar distributions for the deviations in short driving events. However, the *standard* method usually has more deviation at low values due to the $CO_2$ issue mentioned before. The *deconvolution* method is superior for both OFRs: 98% of all OFR

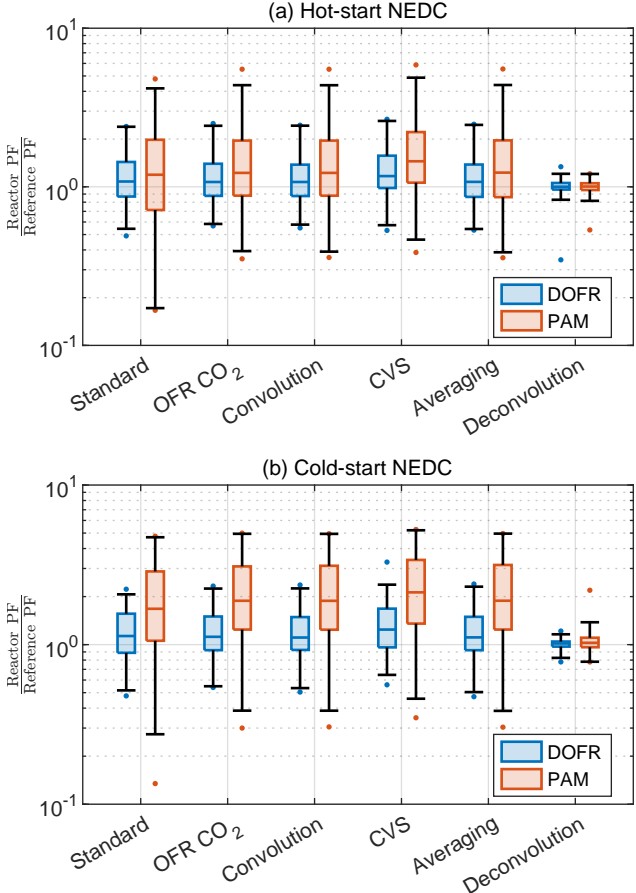

**Figure 6.** The deviation of OFR PFs for 14 s bins in the driving cycles when using different data analysis and measurement methods. Boxes represent 25th and 75th percentiles, and the line inside the box represents the median value. The points are considered outliers if they are greater than 99th percentile or lower than 1st percentile. The whiskers extend to most extreme data points that are not outliers. The PFs are calculated from semi-synthetic SOA data that is linearly proportional to the measured HC concentration in the tailpipe or the simulated HC at OFR outlet.

data is within factors of 0.81 and 1.39 of reference PF. Because of this high accuracy, the applicability of *deconvolution* method in real-world scenarios should be studied in a future publication.

Both reactors tend to overestimate the SOA PFs of short events. For example, in *OFR $CO_2$* method the median ratios between OFR PFs and reference PFs are 1.08 and 1.13 for DOFR in hot- and cold-start cycles, respectively. For PAM, the median ratios are 1.24 and 1.87 in hot- and cold-start cycles, respectively, and in cold-start NEDC 75% of PAM PFs exceed the reference PFs.

## 4.2 Special cases

Although the DOFR usually reports PFs closer to reference values than PAM, this is not always the case. Figure 7 shows two synthetic examples: one where the reference SOA concentration increases simultaneously with exhaust flow rate (typical acceleration observed in the driving cycles presented), and another where the peak in exhaust flow rate is not aligned with the reference SOA peak (e.g., a SOA peak originating from engine start followed by elevated exhaust flow rate due to acceleration after the engine start).

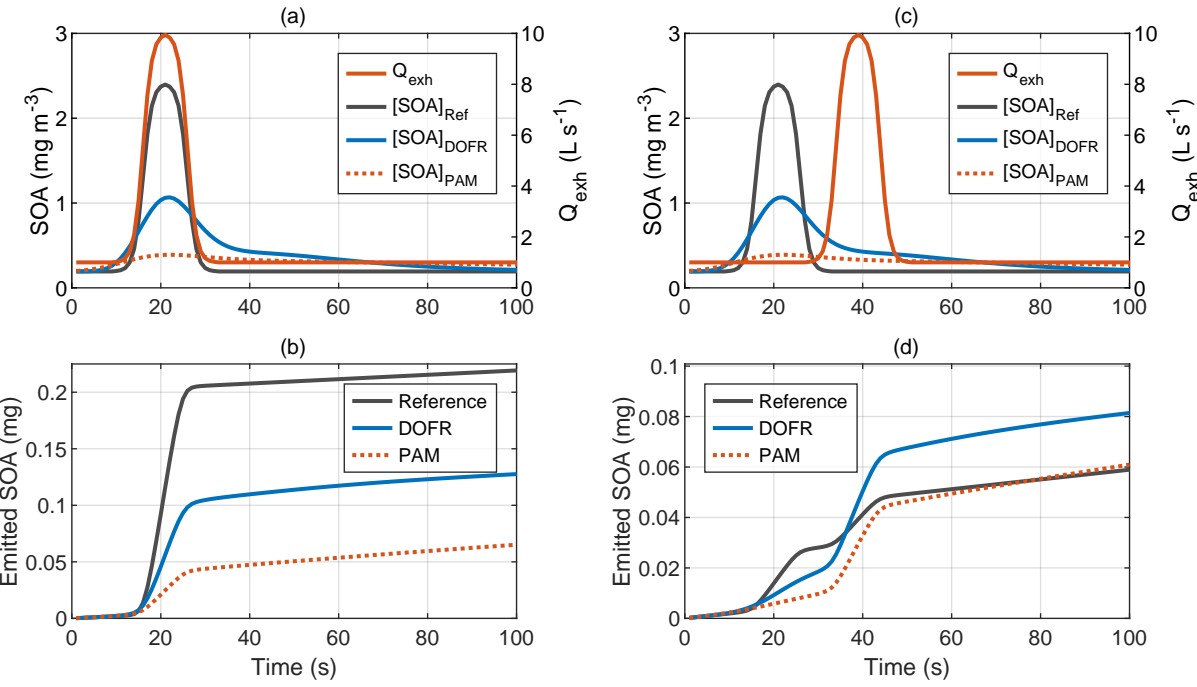

**Figure 7.** Two distinct example time series of exhaust gas concentrations and exhaust flow rate. In the typical acceleration case (a)-(b), the DOFR SOA emission is closer to reference, whereas in the case where reference SOA peak and exhaust flow rate peak are not aligned (c)-(d), the PAM outcome agrees better with reference emission.

In the case where exhaust flow rate and reference SOA concentration peaks are well aligned (Fig. 7a-b), both OFRs report too low SOA emission, which was also the case in Fig. 1, but DOFR result is closer to the reference. However, when the two reference signals are not aligned (Fig. 7c-d), DOFR results in higher overestimation of the emitted SOA because the SOA concentration at DOFR outlet is still elevated when the exhaust flow rate starts to increase. This is the case with PAM as well, but since the SOA peak at PAM outlet is distributed over a longer time period, the concentration is not as high as in DOFR and 420 the resulting SOA emission agrees better with the reference emission.

In Sect. 4.1, the performance of the OFRs was investigated only for one real gasoline vehicle running two cycles, and DOFR typically resulted in better agreement with reference SOA PF than PAM. However, as illustrated in Fig. 7, DOFR does not

result in better agreement in all arbitrary cases. Different vehicle types and more aggressive driving cycles may exhibit different behaviour in tailpipe gas concentrations and exhaust flow rate compared to the gasoline vehicle driving the NEDC, and also the alignment between the concentration peaks and the changes in exhaust flow rate may be different. For example, in Diesel vehicles the $CO_2$ concentration is load-dependent whereas in the gasoline vehicle studied here the tailpipe $CO_2$ concentration was almost constant. Hybrid vehicles may repeatedly switch the combustion engine off and on during the driving cycle.

Thus, to investigate the performance of the OFRs and data analysis methods in a broader range of instances, we performed a Monte Carlo analysis on synthetic driving cycles that include various different combinations of exhaust flow rate, $CO_2$ concentrations and HC concentrations.

### 4.3  Synthetic driving cycles

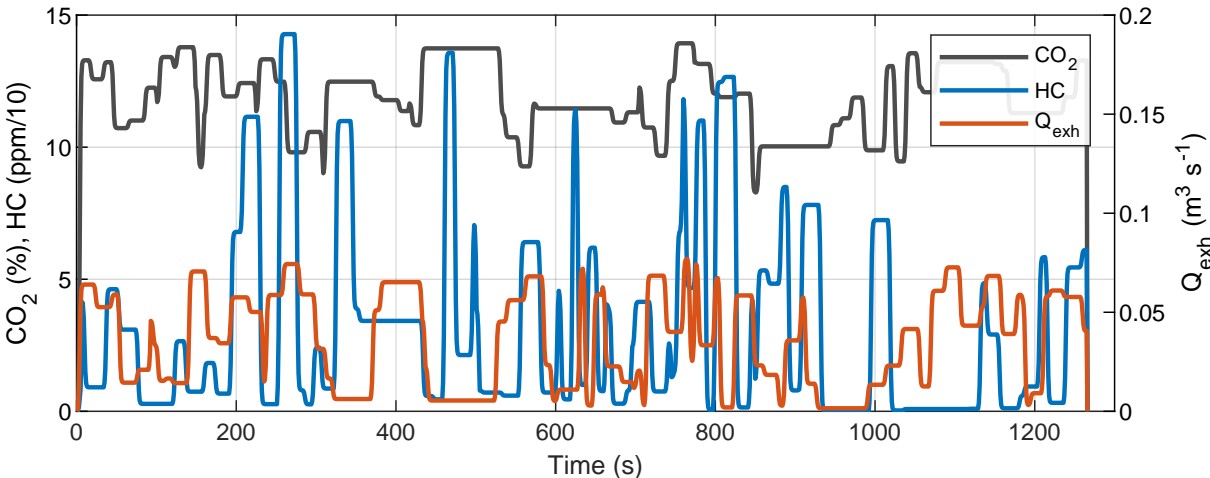

**Figure 8.** An example of a synthetically generated driving cycle.

Driving cycles have three variables that affect the SOA PF: $CO_2$ concentration, HC concentration and exhaust flow rate. The synthetic driving cycles were generated by dividing the cycle in periods of random duration for each variable, where the value of the variable was random (but constant for the period duration). The periods for each variable were generated independently of each other, so that the changes in the values of each variable do not necessarily coincide with changes in the other two variables. The generation algorithm is described in more detail in Sect. S4 and an example of a generated cycle is shown in Fig. 8. More examples are shown in Fig. S17. In total, 10000 synthetic driving cycles were generated.

Figures 9a-b show that the distribution of full cycle SOA PFs was skewed towards underestimation for both reactors, but more severely for PAM, when using the *standard* method for the synthetic driving cycles. The two other methods shown, the *OFR $CO_2$* and *convolution* methods, agreed well with the reference PF. Only three methods are shown here because it was already observed in Sect. 4.1.1 that the *averaging* method is not suitable for calculating the full cycle PFs, and that the *CVS*

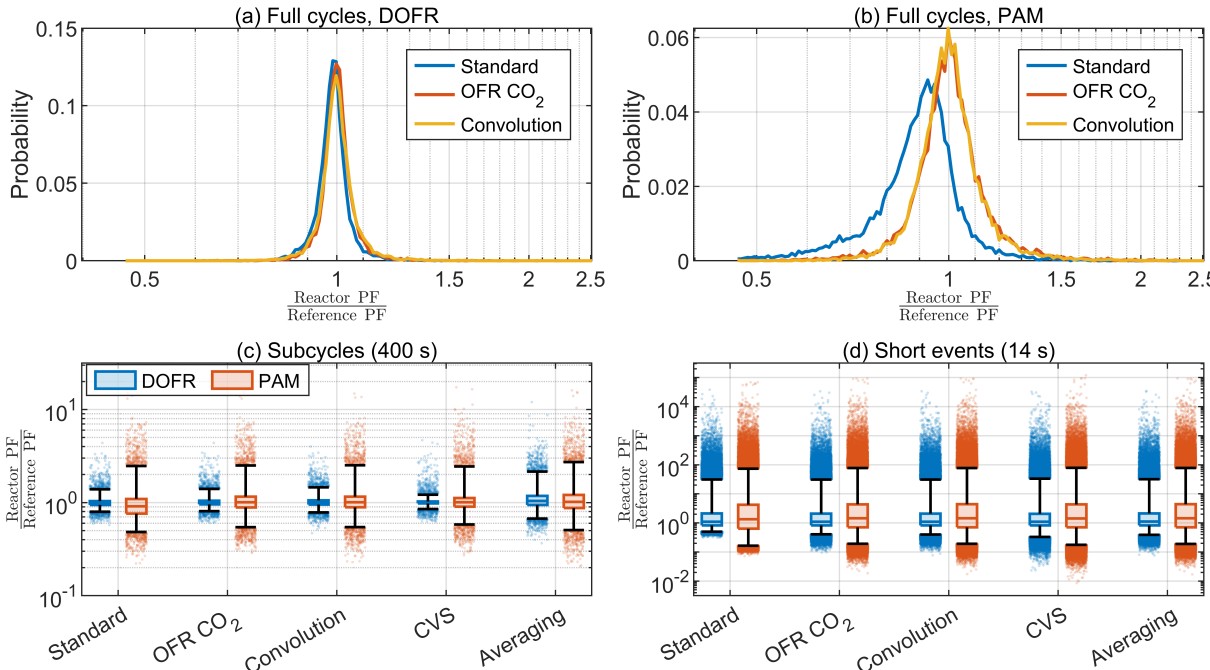

**Figure 9.** The probability distributions for full cycle SOA PFs for 10000 synthetic driving cycles using DOFR (a) and PAM (b), and the the deviation in OFR PFs for 400 s (c) and 14 s bins (d) in the synthetic driving cycles. Boxes represent 25th and 75th percentiles, and the line inside the box represents the median value. The points are considered outliers if they are greater than 99th percentile or lower than 1st percentile. The whiskers extend to most extreme data points that are not outliers.

method always leads to correct full cycle PF. The *deconvolution* method was too time consuming to apply for all 10000 driving cycles.

Similar to full cycle PFs, the standard method typically underestimated the PF for PAM when calculating the PFs for 400 s
subcycles (Fig. 9c). Among the other calculation methods, the *averaging* method led to broadest distribution of deviations and the *CVS* method performed best for both OFRs.

Because of the other disadvantages of the CVS sampling discussed in Sect. 4.1.1, the *OFR $CO_2$* and *convolution* methods seem most feasible for exhaust measurements based on the distributions in Fig. 9c. Using the *OFR $CO_2$* method for 400 s subcycles in the synthetic cycles, the median OFR-to-reference ratio was 1.00 for DOFR and 1.01 for PAM. 50% of DOFR PF
ratios ranged between 0.96 and 1.05 and PAM ratios between 0.89 and 1.16. 98% of DOFR PF ratios ranged between 0.81 and 1.40 and PAM ratios between 0.54 and 2.50.

Figure 9d shows that the different calculation methods resulted in very similar distributions when calculating the SOA PFs for short events. The *CVS* method led to smallest lower outliers for both OFRs, and the lower outliers in *standard* method are closest to 1. The PAM median was closest to reference value when using the *standard* method, but on the other hand the 25th
percentile was smaller than with the other methods. In *OFR $CO_2$* method, 50% of DOFR PF ratios ranged between 0.82 and

2.09 and PAM ratios between 0.71 and 4.38. 98% of DOFR PF ratios ranged between 0.41 and 31.00 and PAM ratios between 0.19 and 77.26. A summary of deviations of OFR-derived PFs from the reference values in both real cycles and synthetic cycles is shown in Table 1.

**Table 1.** Ratios of reactor PF to reference PF when using OFR $CO_2$ method. For each case, the median ratio and 25th, 75th, 1st and 99th percentiles of ratios are shown (notation of $P_{25}$ for 25th percentile etc. is used).

| | | Subcycles (400 s) | | | | | Short events (14 s) | | | | |
|---|---|---|---|---|---|---|---|---|---|---|---|
| | | Median | $P_{25}$ | $P_{75}$ | $P_{01}$ | $P_{99}$ | Median | $P_{25}$ | $P_{75}$ | $P_{01}$ | $P_{99}$ |
| Real cycles | PAM | 1.04 | 0.88 | 1.20 | 0.87 | 1.22 | 1.47 | 1.03 | 2.61 | 0.36 | 4.99 |
| | DOFR | 0.95 | 0.92 | 0.98 | 0.91 | 1.01 | 1.09 | 0.89 | 1.43 | 0.55 | 2.41 |
| Synthetic cycles | PAM | 1.01 | 0.89 | 1.16 | 0.54 | 2.50 | 1.45 | 0.71 | 4.38 | 0.19 | 77.26 |
| | DOFR | 1.00 | 0.96 | 1.05 | 0.81 | 1.40 | 1.11 | 0.82 | 2.09 | 0.41 | 31.00 |

Since a significant fraction of OFR-derived PFs were more than 10-fold compared to reference PFs for short, 14 s segments, and the agreement was better for longer (400 s) segments, it is of interest to determine what is the minimum segment durations for which the OFR results are accurate enough. If we choose that the OFR result is acceptable when 99% of the PFs are less than three times the reference PF, the minimum duration was 110 s for DOFR and 350 s for PAM. The deviations as a function of bin duration are shown in Fig. S12.

## 5   Conclusions

In this study, the effect of OFR transfer function on the accuracy of SOA PFs in transient driving cycles was investigated by using semi-synthetic and synthetic exhaust gas data. The analysis was done for two OFRs: a PAM reactor with a broad transfer function and DOFR with a narrower transfer function.

Even though the wide residence time distributions of OFRs resulted in momentary PFs that differed from the reference PFs, it was possible to determine the integrated PFs relatively accurately for longer periods within the driving cycles. However, a wrong data analysis method could lead to PFs that were more than double of reference PF.

When determining SOA PFs of short-duration events in a driving cycle, such as accelerations, the errors were larger for both OFRs compared to PFs of longer periods. The narrower transfer function of DOFR was advantageous, since the 1st and 99th percentiles of OFR-to-reference PF ratios in the short-duration events (14 s periods) in real driving cycles were 0.55 and 2.41, respectively, for DOFR and 0.36 and 4.99 for PAM (using *OFR $CO_2$* method).

When extending the study to synthetic driving cycles, the OFRs still reported the full cycle PFs with relatively small error. For 14 s bins in the synthetic driving cycles, it was observed that the OFRs may overestimate the SOA PFs by more than factor of ten. It is questionable whether any of the studied OFRs can be used to determine SOA PFs for that short driving events since the potential error is so high. At least, the uncertainty should be addressed when reporting the PFs. On the other hand, the synthetic driving cycles are random and do not necessarily represent typical driving cycles, so the results represent maximum

possible error rather than typical error observed in real driving cycles. More real cycles should be studied in order to evaluate the potential error.

By lengthening the bin duration to 110 s, the 99th percentile of DOFR PF ratios was below 3 in the synthetic driving cycles. The 99th percentile of PAM PF ratios was below 3 when the bin duration was longer than 350 s.

Deconvolution of SOA concentration measured at OFR outlet seemed promising method because it enhanced the accuracy of SOA PFs significantly. However, the result was obtained by assuming noise-free measurement of mass concentration, and thus its applicability to real-world scenarios should be further studied.

There are also other reasons than the transfer function alone for OFRs to report incorrect SOA production factors, such as non-tropospheric gas chemistry or non-tropospheric losses. In this paper, we only studied the error that is caused by the OFR transfer function. Other sources of error were isolated by assuming that the oxidation in OFRs perfectly reproduces atmospheric oxidation, that there are no non-atmospheric losses in the OFRs, and that there are no absorption or adsorption related delays in the OFR. The analysis is limited to conditions where SOA formation potential is directly proportional to HC concentration and where the proportionality is constant throughout the driving cycle, i.e., the OH exposure in the OFR is sufficient to oxidize all precursor gases completely.

Even though the assumption of SOA concentration being directly proportional to HC concentration does not generally hold, the HC measurement from tailpipe accompanied with the methods presented in this study is a good sensitivity test for transfer function -related uncertainties when determining the SOA PFs with an oxidation flow reactor. Similar analysis apply when using any other slow-response instrument to determine emission factors.

Arising from our analysis, we present the following best practise recommendations for OFR emission measurements:

– Before the start of the cycle, the reactor must be sampling zero air to avoid previous driving affecting the cycle SOA PF. The exhaust sampling must start at the same time as driving cycle starts. This concerns the engine-off periods of hybrid vehicles as well: zero air sampling should be started immediately when the combustion engine is switched off, and the tailpipe sampling started when the engine turns back on. When sampling from CVS, this is done automatically.

– When the cycle ends, the reactor must immediately start sampling zero air. The measurement must be continued at least for duration of the OFR $\tau_{peak}$ to make the delay correction in data after-treatment possible. When sampling from CVS (or when using the convolution method), the sampling of zero air must be continued at least for duration of OFR mean residence time, but longer sampling time will result in more accurate PF (Fig. S13).

– In order to use the OFR $CO_2$ method or convolution method, $CO_2$ should be measured downstream of the OFR, or the OFR outlet $CO_2$ concentration should be simulated by convolving the tailpipe concentration with the OFR transfer function and dividing with the dilution ratio.

– When using other than *standard* or *deconvolution* method, the distance-based production factors should be calculated by first calculating the fuel-based production factor with one of the presented methods, and then using OBD data to convert the fuel-based PF to distance-based.

The Matlab code used in this study is available as a Supplement file to reproduce the analysis for any OFR with a known transfer function and for any driving cycle for which the $CO_2$ and HC concentrations and exhaust flow rate are available.

*Code and data availability.*  The engine exhaust data for the real driving cycles is available in the Supplement. The Matlab code to reproduce the analysis is available in the Supplement.

## Appendix A: Experimental details

### A1  OFR characterization

Dekati oxidation flow reactor (DOFR) is a commercial oxidation flow reactor, which dimensions are very close to those of
Tampere secondary aerosol reactor (TSAR; Simonen et al. (2017)). The main geometrical additions compared to TSAR are a conical outlet, a laminating grid element in the inlet and unlike TSAR, all sample is evacuated through a single outlet. The oxidation reactor is surrounded by 12 UV lamps of which two can be switched on individually and the rest of the lamps in pairs, whereas TSAR has two intensity-controlled UV lamps (Kuittinen et al., 2021a). The housing of the oxidation reactor is cooled with air. The air cooling in the commercial version is enhanced compared to the prototype version used here. Similar to
TSAR, DOFR is an OFR254 type reactor, which means that OH radicals inside the reactor are generated by 254 nm UV light from externally mixed ozone and water vapor.

The transfer function of DOFR was determined for $CO_2$ and toluene by injecting 10 s square pulses of gases into the reactor and measuring them downstream of the reactor. The $CO_2$ was measured with LI-840 analyzer (LI-COR Inc.) and toluene with Vocus proton transfer reaction mass spectrometer (Aerodyne Research Inc.). The gases were injected at the enclosure inlet and
$CO_2$ was measured directly downstream of the reactor while the toluene was measured downstream of ejector diluter, which is an integral part of DOFR. Thus, the toluene RTD describes the response of the full unit, although we assume that this is the case for $CO_2$ RTD as well because the residence time in the diluter and its sampling lines is minor. The mean flow rate through DOFR was 6.8 lpm during the $CO_2$ experiments and 6.0 lpm during the toluene experiments.

The square pulses were generated by continuously injecting constant mass flow rate of $CO_2$ or $N_2$ mixed with toluene into
a fast pneumatic 3-way valve (MS-151-DA actuator with SS-42GXS6MM-51D 3-way valve; Swagelok Company), one outlet connected to the DOFR inlet and the other to the excess line. The toluene vapor was generated with a permeation oven (V-OVG; Owlstone Inc.). The measurement setup is shown in Fig. S2.

The DOFR RTDs of 10 s pulses were measured for 3 different UV lamp configurations: 'off', 'low' (two central UV lamps on), and 'high' (all UV lamps on). The $O_3$ generation was switched off to prevent toluene reacting with OH radicals
when measuring the toluene RTD. The measured RTDs correspond to 10 s input pulses, so they do not represent the actual transfer function which is the response to a Dirac delta input. Thus, the OFR transfer functions were determined by finding the transfer function that resulted in best agreement with the measured concentration when convolving with 10 s square pulse. The candidate function was a linear combination of Taylor distributions (Lambe et al., 2011; Huang and Seinfeld, 2019), and

the best fit was found with Matlab function *'fit'*. The gas analyzer response was not determined separately, so it is included in the reported transfer functions. In this study, the transfer function corresponding to 'low' UV lamp configuration was used to simulate the DOFR output. This lamp configuration resulted in OH exposure of $7.9 \cdot 10^{11}$ cm$^{-3}$ s$^{-1}$ according to toluene measurements. The DOFR transfer functions for $CO_2$ and toluene are shown in Fig. S5, and the comparisons between the convolved square pulses and the measured DOFR output concentrations are shown in Figs. S6 and S7.

By switching the $O_3$ reactor on, we also measured the mass concentration that was produced from 10 s toluene pulse for the 'low' UV lamp configuration. The mass concentration was measured with an electrical low-pressure impactor (ELPI, Dekati Ltd.; Keskinen et al. (1992)) with improved nanoparticle resolution (Yli-Ojanperä et al., 2010). It would be possible to determine a transfer function for SOA formation based on these measurements, but since such data was not available for PAM chamber, we simulated the SOA formation in both OFRs by assuming that the SOA formation response is equal to $CO_2$ response. Simonen et al. (2017) did measure the PAM SOA formation for a toluene pulse, but in those measurements the PAM ring flow was not used. Since the usage of ring flow is a standard method in PAM measurements and affects the transfer function, we used the $CO_2$ pulse data measured by Lambe et al. (2011) to determine the PAM transfer function by the same fitting procedure as for the DOFR (Fig. S8). In the measurements by Lambe et al. (2011), PAM ring flow was used and the UV lamps were on. Using the $CO_2$ transfer function to simulate the SOA formation in DOFR resulted in a satisfactory agreement with the experimental data (Fig. S1a), so the usage of $CO_2$ transfer function in this study is justified.

For the OFR delay correction (Eq. 8) we used the peak residence time as the correction constant. The peak residence time ($\tau_{peak}$) is the residence time for maximum value in the transfer function (i.e., $E(\tau_{peak}) = \max(E(t))$). Figure S10 shows that the error in SOA PF was smallest when the delay correction constant was close to $\tau_{peak}$.

**A2    Vehicle exhaust measurements**

The vehicle in real driving cycle measurements was a Euro 6 gasoline vehicle equipped with 1.4 l turbocharged direct injection engine (110 kW). The vehicle was soaked for 15 h before the cold-start cycle and pre-conditioned by driving at 80 kmh$^{-1}$ for 5 min before the hot-start cycle. The hot start cycle started with idling engine. In the simulations, it was assumed that the OFRs are flushed with zero air until the cycle starts and immediately after the cycle ends. So even though engine is running before the start of hot-start NEDC, the OFRs are filled with zero air at $t = 0$ s.

The total hydrocarbon concentration (methane equivalent ppm) was measured with a flame ionization detector and the $CO_2$ concentration with a non-dispersive infrared analyzer. Both gases were sampled directly from tailpipe. The exhaust mass flow rate was calculated based on the intake air flow rate and fuel consumption obtained from the on-board diagnostics data. The fuel carbon content ($k'$) of 860 g kg$^{-1}$ was used in the calculations.

*Author contributions.* P.S.: Original idea, data processing, writing the manuscript, planning and execution of experiments. M.D.M.: Manuscript conceptualisation and preparation. P.P.: DOFR experiments, Vocus data processing, commenting the manuscript. A.H., A.K.: DOFR experiments, commenting the manuscript. P.M.: Planning of chassis dynamometer experiments and commenting manuscript. P.K.: Project man-

agement, planning and execution of chassis dynamometer experiments and commenting manuscript. J.K.: Funding, project management, manuscript conceptualization and preparation.

*Competing interests.* J.K. is a member of the board of Dekati Ltd.

*Acknowledgements.* The chassis dynamometer measurements were conducted within Health relevant and energy efficient regulation of
exhaust particle emissions (HERE) project [decision number 40330/13] funded by Business Finland (Tekes). We acknowledge additional funding from the Academy (now: Research council) of Finland's Flagship programme (decision No.'s 337551, 357903) and infrastructure funding (decision No.'s 328823, 345528). We acknowledge the engine laboratory personnel at Dinex Finland Oy for the chassis dynamometer measurements and Dekati Ltd for providing a DOFR prototype for testing.

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
