# Peer review of "Estimating errors in vehicle secondary aerosol production factors due to oxidation flow reactor response time"

_EGUsphere, 2023_

## Author Comment (AC1)

**Response to referees**

We thank both reviewers for taking their time and giving constructive comments and suggestions. We give the point-by-point responses to all comments below in blue text, and the original referee comments are in black text. In addition, we attach the revised versions of the manuscript and the supplement at the end of this document, where all the modifications are marked.

Based on the suggestions from both referees, we added a "Methods" section between the sections "Theoretical background" and "Results and discussion". As a consequence, some of text from Sections 2 and 5 is moved to the new Section 3 in the revised manuscript, and the approach in Section 2 is simplified to improve the flow of the text. The detailed information of the experimental methods (Section 5 in the original manuscript) is moved to Appendix. The assumptions regarding the SOA formation and the validity of the assumptions are now discussed in Section 3 in the revised manuscript.

Apart from the changes suggested by the referees, we improved the readability of the figures as follows: instead of showing the logarithm of the ratio between OFR SOA PF and reference PF, we show the actual ratio and use logarithmic axis instead of linear.

On behalf of all authors,

Pauli Simonen

**Response to Anonymous Referee #1**

The manuscript by Siomnen et al. discusses the influence of residence time distribution (RTD) of oxidation flow reactor (OFR) on SOA formation from car exhaust from some testing driving modes. They derived the equation for representing temporal variation in SOA concentration following OFR. Further analysis of experimental and synthetic data suggested that numerical deconvolution for the influence of VOC emission and RTD of OFR for observed temporal variation in SOA mass is needed, as the time scale for change in driving conditions is shorter than a residence time of typical OFRs. The topic is within the scope of the interest of readers of the journal. Some implicit assumptions in the manuscript will need to be clarified. Organization of the manuscript can be improved. In my opinion, the current quality of the manuscript does not meet with the criteria of the journal, even though the concept of the manuscript itself is interesting.

**Major comments**

**1. Influence of VOC oxidation kinetics.**

Not all the VOCs would be consumed in the reactor if their oxidation time scale is longer than the mean residence time of the OFR. I wonder if exponential decay in VOC concentration ([VOC] = [VOC]0 * exp (-t/tau)) in the OFR should also be considered for deconvoluting the data. If the oxidation time scale in OFR is sufficiently short due to the high concentration of oxidants, it will need to be quantitatively discussed.

The anthropogenic SOA precursors present in vehicle exhaust are mainly reactive towards OH radicals. If the average OH exposure in the OFR is approximately 6 equivalent days in the atmosphere, most identified anthropogenic SOA precursors are almost completely consumed in the OFR even at the shortest residence times, but some exceptions exist, such as benzene.

This issue is now discussed in Sect. 3 of the revised manuscript and in more detail in Supplement Sect. S2.

**2. Equation (5)**

The equation (SOA = Y·HC) implies that all hydrocarbons in the OFR are oxidized. My understanding is that HC indicates the total amount of injected hydrocarbons, rather than reacted amount of hydrocarbons. Could the authors provide a justification for this assumption?

The equation also assumes that SOA yield does not change throughout the experiment, even though both gas phase chemical composition and SOA mass concentration in the OFR keeps changing. I personally think that Y should also be a function of t. Could the authors provide future detailed information/discussion about it?

The assumption that all hydrocarbons in the OFR are oxidized is now discussed in Sect. 3 in the revised manuscript and in Sect. S2 in the revised Supplement. A new figure (S1) is added in Sect. S2, and the contents in Fig. S8 were moved to Fig. S1a in the revised Supplement.

On the other hand, the SOA in Eq. (5) is the reference SOA, i.e. the maximum potential SOA that could be formed in the atmosphere, and thus the assumption that all the precursors are oxidized is justified. This was not clear in the original manuscript.

The approach in the Theoretical background is now clarified so that we only introduce the term "reference SOA" ($[SOA]_{ref}$) in Sect. 2. This can be any time series of SOA formation potential that has temporal variation. Later, in the new Sect. 3 in the revised manuscript, we define a SOA reference that is directly proportional to HC concentration. Even though this is not totally realistic since the VOC composition in the exhaust probably changes within the driving cycle, we believe that the temporal variation in the HC concentration still reflects the temporal variation in the SOA formation potential of vehicle exhaust better than fully arbitrary SOA reference.

However, as the Referee points out, even though the VOC composition was constant in the exhaust and Y was constant, the SOA yield in the OFR may change between different parts of the driving cycle because of e.g. changes in organic aerosol mass concentration. This is now discussed in Sect. 3 in the revised manuscript.

**3. Discuss adsorption/absorption of VOCs/OVOCs on the wall**

In most cases, the concept of RTD is employed by assuming that a flow pattern of fluid is the dominant regulator for determining the time scale for reactants to stay in a chemical reactor. However, in the case of VOCs/OVOCs for SOA precursors, absorption/desorption processes on walls of reactors are typically non-negligible for determining their actual residence time. This process slows down the response of a chemical reactor to the changes in operating conditions. It will be helpful if the authors could provide how the process influences responses of the OFR.

This is an important aspect that was not considered in the original manuscript. The delays caused by adsorption or absorption could severely affect the alignment between the exhaust flow rate and the SOA formation not only because of the reactor walls but also because of the same phenomena in the sampling lines between the tailpipe and the OFR.

The effect of adsorption/absorption is discussed in Sect. 3 in the revised manuscript with relevant citations. We could not find parametrization for delays of typical SOA precursors in the literature. Based on our experiments with DOFR, such delays are minor for toluene SOA formation. However, based on

the literature, the delays are longer for less volatile organic gases and could be thus relevant for intermediate volatile organic compounds present in Diesel vehicle exhaust.

**4. Method section.**

It is better to put the method section prior to the result section, as in the case of most of other publications in the journal. The manuscript indicates that some data sets were obtained using the constant volume sampler (CSV). The present description about the CSV is not sufficient to understand how the CSV works/why it is needed/what are the advantages and disadvantages to use it.

As described earlier, we added a Methods section prior to the Results section according to the suggestion.

We added background information on why the CVS is used in regulated emission measurements and improved the description in Sect. 2.2 in the revised manuscript. The disadvantages of CVS in OFR measurements are already discussed in lines 267-279 in the original manuscript.

**5. Section S3. Synthetic driving cycle**

Consider moving this section to the main text, as it is critical information for understanding the contents.

We feel the detailed list in Sect. S3 (in original Supplement) would be exhausting in the main text and would not improve the flow of the text. A summary of the generation of a synthetic driving cycle is provided in the beginning of Sect. 3.3 in the original manuscript. We added an example figure of a synthetically generated driving cycle in this section in the revised manuscript to assist understanding the contents.

**Minor comments**

**6.1:** Line 23: What does m-% mean?

The notation of mass percentage is changed to a clearer expression: **"A mass fraction of 20-62% of these emissions...".**

**6.2:** Figure 1: In this figure (and at many other parts in the manuscript), parameter t is employed for two meanings. One is the time after the start of a driving test, and another is time for residence time in the OFR. However, these two types of t do not correspond to each other except for the case of pulse injection. I suggest the authors to consider using different parameters for clarification.

We agree that the residence time distribution shown in Fig. 1 can be confusing. To clarify Fig. 1, we removed the graph showing the RTD and replaced it with text to indicate that the HC measured at the OFR outlet is affected by the RTD.

We changed the delay correction factor in Eq. (7) from $\tau$ to $\tau_r$. In order to not mix the $\tau$ in Eq. (7) with the $\tau$ in Eq. (2), we changed the variable of integration in Eq. (2) from $\tau$ to $\gamma$. The parameter $t_{mean}$ in Fig. S13 is changed to $\tau_{mean}$.

**6.3:** Line 117: I could not understand how the delay caused by the OFR can be calculated.

We clarified this section and defined how the peak residence time is calculated.

**6.4:** Line 129: The concepts of [SOA]OFR and [HC]OFR are clear to me. However, I am wondering how these two metrics can simultaneously be measured during practical applications.

As described in previous responses, we clarified Sect. 2 so that the notation [HC]OFR is no longer used (although it is still used in the Introduction). However, if one is interested in the correlation between the exhaust HC concentration and the SOA concentration measured downstream the OFR, there are three alternative methods to determine [HC]OFR (the HC concentration downstream the OFR when the oxidation in OFR is disabled):

1) Measure the HC concentration upstream of the OFR and convolve the concentration with OFR transfer function.
2) Use two identical OFRs in parallel, so that the oxidation is disabled in the other and measure the HC concentration from the non-oxidizing OFR outlet.
3) If the driving cycle is repeatable, one could measure two repetitions of driving cycles and disable the OFR oxidation in the second one.

After this, it is possible to make a correlation plot between [SOA]OFR and [HC]OFR.

**6.5:** Line 172: What does 'transfer function standard deviation' mean? Does it indicate width of the RTD? If so I wonder how the standard deviation was derived, as functional forms of RTD are not normal functions in many cases.

It is correct that the standard deviation was reported to indicate the broadness of the transfer function.

The transfer function is analogous to a probability density function, so its standard deviation ($\sigma$) is defined as:

$$\sigma = \sqrt{\int_0^\infty (t - \tau_{mean})^2 E(t)\, dt}\,,$$

where $\tau_{mean}$ is the mean residence time of the OFR: $\tau_{mean} = \int_0^\infty t E(t)\, dt$.

The Referee is correct that it is challenging to obtain analytical solution for the standard deviation when using Eq. S1 to define the transfer functions. Thus, both integrations were performed numerically by first calculating the numerical values of E(t).

The information about standard deviation calculation is added to Supplement Sect. S1, and the following text is added to the manuscript:

**"The standard deviation of the transfer function reflects the transfer function broadness and its calculation is presented in Sect. S1."**

**6.6:** L 188: I do not think that using the expression of 'true SOA' is very appropriate. The value was estimated from measured hydrocarbon concentration by assuming that SOA yield is always a constant. However, the validity of this assumption is unclear. It is not a good idea to use such an expression unless there is convincing evidence about how the true SOA mass should be.

We agree that the term 'true SOA' can be misleading. Thus, when referring to SOA concentration or production factor that was calculated based on HC concentration, we use the term 'reference SOA' instead in the revised version of the manuscript. The term is replaced in all necessary figures.

**Response to Anonymous Referee #2**

The paper by Pauli Simonen et al. presents an insightful exploration into the complexities of measuring vehicle secondary organic aerosol (SOA) production factors (PF) using oxidation flow reactors (OFRs). The authors' approach, particularly their consideration of OFR response time and its convolution with vehicle emissions during transient driving cycles, provides valuable insights into determining accurate PFs. By utilizing synthetic and semi-synthetic exhaust emission data, this study not only evaluates potential errors but also suggests methods for their mitigation, emphasizing the importance of constant volume sampling (CVS) for precise measurements. The research importantly points out the need for a thorough understanding of OFR transfer functions and response times in aerosol research. Before recommending acceptance of the paper, I have the following suggestions for the authors:

1. The paper defines the concept of SOA yield (Y) as the ratio of produced SOA to consumed hydrocarbon (HC). However, in the derivation process, "total" HC is used instead of "consumed" HC to calculate the SOA produced in OFR. Equation 5 implies a default assumption that all HC is completely oxidized in the OFR, which may not always be accurate in real scenarios.

Please see the response to Referee #1 comments 1 and 2.

2. In the absence of any sink within the OFR (such as wall loss or chemical reactions), the cumulative emissions measured before and after the OFR should match over time, since the CVS merely dilutes emissions without exhausting them. If this is not the case, the authors should present a clear mass balance scheme to explain the differences observed before and after the OFR.

The convolution used to simulate the OFR outlet concentrations does not include any sinks. Thus, the definite integral the concentration simulated at the OFR outlet equals the definite integral of the concentration at the OFR inlet, given that the upper boundary of integration extends long enough after the concentration at the inlet has reached zero.

When using the CVS, the same is true for the emission, when the integral is multiplied with the CVS total constant volumetric flow. The proof of this is shown in Eqs. (11)-(14) in the original manuscript. It is also shown in Fig. S13 in the original Supplement that when the OFR is sampling from CVS, the SOA PF calculated based on the OFR SOA concentration approaches the reference SOA PF when the integration upper boundary approaches infinity after the driving cycle ends (and the precursor concentrations reach zero in the CVS).

When the lower boundary of integration is not zero or the upper boundary does not approach infinity, the proof is not applicable, and the CVS sampling does not necessarily result in correct PF. This can be seen e.g. in Fig. 8 in the original manuscript, where the error in CVS method is similar to the error in the other methods where direct sampling from tailpipe is used.

3. Equation 14 in the paper derives the total cumulative SOA, but the methodology for deriving SOA(t) is not clearly explained. Clarification on this derivation would enhance the readers' understanding.

The approach in Sect. 2 was clarified in the revised manuscript. As suggested by the Referee, we unambiguously defined the time-dependent concentration of reference SOA ($[SOA]_{ref}$) in the revised manuscript Eq. (15), and the concentration of SOA measured at the OFR outlet ($[SOA]_{OFR}$) in the revised manuscript Eq. (6).

4. There is ambiguity between [HC]'OFR and [HC]OFR as mentioned in Lines 128-129.

The concept of $[HC]_{OFR}$ is removed from the revised manuscript in Sect. 2 as it is a quantity that is not normally measured. Instead, we only define the $[SOA]_{OFR}$ in the revised manuscript. The difference between $[SOA]_{OFR}$ and $[SOA]'_{OFR}$ is that the latter is corrected for the "average" delay in the OFR to better align the measured SOA concentration at the OFR outlet with exhaust flow rate or other concentrations measured directly from the tailpipe.

5. The structure of the paper could be improved for better flow and coherence. Specifically, Section 5, which discusses Methods, should be relocated to an earlier part of the manuscript to enhance the logical progression of the paper.

As discussed in previous responses, we re-organized the Sections 2 and 5 into Sections 2 and 3, where the latter discusses the experimental and computational methods used in the analysis. We believe this improves both understanding the theoretical background and the approach used in this study, and the structure and flow in general. Since Section 5 in the original manuscript was partially very detailed, we do not include all the text in the new Methods section but move the text into new Appendix.

[revised manuscript text omitted]

**S1  OFR transfer functions**

The OFR transfer functions were defined as linear combinations of Taylor distributions:

$$E(t) = \sum_i \frac{f_i}{2} \exp\left(-\frac{\mathrm{Pe}_i(\tau_i - t)^2}{4\tau_i t}\right)\sqrt{\frac{\mathrm{Pe}_i}{\pi \tau_i t}}, \tag{S1}$$

where

$$\sum_i f_i = 1. \tag{S2}$$

We used a combination of two Taylor distributions for DOFR and three for PAM. The parameters for calculating the transfer functions for different OFRs are shown in Table S1.

The standard deviation of the transfer function reflects the transfer function broadness. Because the transfer function is analogous to a probability density function, its standard deviation ($\sigma$) is defined as

$$\sigma = \sqrt{\int_0^\infty (t - \tau_{mean})^2 E(t)dt}, \tag{S3}$$

where $\tau_{mean}$ is the mean residence time defined as

$$\tau_{mean} = \int_0^\infty t E(t)dt. \tag{S4}$$

The standard deviation was calculated numerically by first calculating the values of the transfer function with Eq. S1, then calculating the mean residence time with Eq. S4 and finally obtaining the standard deviation with Eq. S3. The upper limit of integration was 700 s.

**S2    Effect of chemical kinetics on SOA response**

In the analysis, it is assumed that all SOA precursors get oxidized in the OFRs (Sect. 3). In other words, the square pulse of any SOA precursor is assumed to produce a SOA mass concentration profile at the OFR outlet that is similar to a $CO_2$ profile produced by a square pulse injection of $CO_2$. This assumption can be written as

$$[SOA]_1(t) = ([VOC]_0 * E)(t) \cdot y, \tag{S5}$$

where $y$ is the SOA yield, $E$ is the transfer function of $CO_2$ and $[VOC]_0$ is the time-dependent precursor (volatile organic compound) concentration at the OFR inlet. A more advanced approach considers that the formation of SOA depends on the amount of precursor gas that is oxidized by OH radicals, as discussed in Sect. 3. This can be written as

$$[SOA]_2(t) = \Delta[VOC](t) \cdot y = (([VOC]_0 * E)(t) - [VOC]_f(t)) \cdot y, \tag{S6}$$

where $[VOC]_f$ is the precursor concentration at OFR outlet. Assuming the precursor reacts only with OH radicals, its concentration at OFR outlet is

$$[VOC]_f(t) = ([VOC]_0 * E)(t) \cdot \exp(-k \cdot [OH]_{avg} \cdot t), \tag{S7}$$

where $k$ is the reaction rate constant between the precursor and OH radical and $[OH]_{avg}$ is the average OH radical concentration in the OFR. Note that Eq. S7 is applicable only for a short input pulse of precursor gas. The average OH radical concentration

is defined as

$$[OH]_{avg} = \frac{OH_{exp}}{\tau_{mean}}, \tag{S8}$$

where $OH_{exp}$ is the average OH exposure in the OFR. The average OH exposure is a measurable quantity that can be determined by measuring the decay of an OH reactive substance in the OFR (e.g. Barmet et al. (2012)). For a steady input of precursor gas, the product $[OH]_{avg} \cdot t$ in Eq. S7 should be replaced with the average OH exposure.

For toluene SOA production in DOFR the basic approach (Eq. S5) led to a satisfactory agreement with the experimental data (Fig. S1a), with very little difference to the more accurate approach (Eq. S6) because almost all toluene is consumed by OH radicals even at the shorter residence times. For PAM (Fig. S1c), the difference between the two simulations is higher because the fraction of sample that spends the least time in the reactor has too low OH exposure to oxidize all toluene. The SOA formation by Eq. S6 in Fig. S1 is calculated by assuming an average OH exposure of $7.9 \cdot 10^{11}$ $cm^{-3}$ $s^{-1}$, since this was

the average OH exposure in the toluene experiment shown in Fig. S1a.

The effect of chemical kinetics is higher for precursor gases that react slower with OH radicals. The reaction rate constant of toluene is $5.63 \cdot 10^{-12}$ $cm^3$ $s^{-1}$, whereas the reaction rate constant of benzene, another common anthropogenic precursor, is smaller, $1.22 \cdot 10^{-12}$ $cm^3$ $s^{-1}$ (Atkinson and Arey, 2003). Figures S1b and S1d show that the assumption in Eq. S5 is invalid for benzene.

[Figure]

**Figure S1.** Measured SOA (downstream of the internal ejector diluter, dilution ratio 8.5) with 'low UV' setting compared to SOA modeled with Eq. S5 ($[SOA]_1$) and with Eq. S6 ($[SOA]_2$) (a), where $[VOC]_0$ is a convolved 10 s square pulse of toluene, and $y$ is the SOA yield determined from steady input experiments. Toluene concentration upstream DOFR during the pulse was 398 ppb, measured from a steady input experiment. The square pulse was convolved with $CO_2$ transfer function. Average flow rate was 6.0 slpm. All concentrations are normalized to the maximum value of $[SOA]_1$. The model results are shown also for benzene SOA formation in DOFR (b) as well as for toluene (c) and benzene (d) in PAM reactor. The OH exposure shown in the right y axis corresponds to the OH exposure experienced by the VOC exiting the reactor at time $t$. Even though the average OH exposure of the sample (VOC + air) measured at OFR outlet is constant, the OH exposure experienced by the VOC is time-dependent in case of pulse injection.

45     The applicability of the assumption in Eq. S5 in general depends on the composition of precursor gases in the vehicle exhaust. According to VOC data presented by Timonen et al. (2017) for gasoline vehicle exhaust in cold-start phase, the average reaction rate constant of the measured aromatic VOCs weighted by their mass concentration was $9.08 \cdot 10^{-12}\ \mathrm{cm^3\,s^{-1}}$. Since this value is higher than the toluene reaction rate constant, we consider the assumption in Eq. S5 sufficient to model SOA formation in

gasoline vehicle exhaust for the purposes of this work, as long as the average OH exposure in the reactor is approximately

50    $7.9 \cdot 10^{11}$ cm$^{-3}$ s$^{-1}$ or higher.

Additionally, all the aromatic VOCs listed by Atkinson and Arey (2003) except benzene, toluene and tert-butylbenzene have higher reaction rate constants than toluene. Regarding the precursor gases in diesel exhaust, all the intermediate volatile organic compounds in diesel vehicle exhaust that were speciated by Zhao et al. (2015) have higher reaction rate constants than toluene.

**S3    Deconvolution**

The deconvolved signal ($[C]^*(t)$) was calculated by using a non-linear programming solver *fmincon* (Matlab R2021b). The solver tries to find the non-negative signal that, when convolved with the OFR transfer function, results in minimal sum of residual squares. In other words, the solver tries to find $[C]^*(t)$, for which $\sum(([C]^* * E)(t) - [C]_{OFR}(t))^2$ is smallest, where $[C]^*(t) \geq 0$ and $[C]_{OFR}(t)$ is the SOA concentration measured downstream of the OFR. For all deconvolution cases presented

60    here, the solver converged to an optimal solution.

**S4    Synthetic driving cycles**

Examples of synthetic driving cycles are shown in Fig. S17. The synthetic driving cycles were generated with the following algorithm:

1) The cycle length is is a random value between 240 s and 2400 s with uniform probability distribution.

65    2) The vehicle type is either Diesel or Gasoline with equal probabilities. The vehicle type affects the behaviour of $CO_2$ concentration in step 2 of $CO_2$ concentration algorithm.

Exhaust flow rate:

1) Choose whether the engine is on or off (can be off when measuring hybrid engine vehicles). The probability for engine off condition is 0.01.

70    2a) If the engine is off, choose a random value between 10 s and 600 s for the duration of engine off period. The exhaust flow rate is zero during the engine off period. Start a new period at the end of this period and define the next period by moving back to step 1.

2b) If the engine is on, choose whether the period of constant value for exhaust flow rate is a stable period (duration between 25 s and 100 s) or a short period (duration between 2 s and 25 s). The probability for a stable period is 0.1 and the

75    probability for duration is uniformly distributed in the specified range.

3) Choose a random number between $0.75 \cdot 10^{-3}$ and 0.08 m$^3$ s$^{-1}$ as the constant value for the exhaust flow rate for this period. The probability is uniformly distributed in this range.

4) Assign the new value for the period by generating a smooth transition between the previous value and the new value with Eq. S9.

80  5) Start a new period following this period and define the next period by moving back to step 1. Repeat the steps until the end of the cycle is reached.

$CO_2$:

1) Choose whether the period of constant value for $CO_2$ concentration is a stable period (duration between 25 s and 100 s) or a short period (duration between 2 s and 25 s). The probability for a stable period is 0.1 and the probability for

85  duration is uniformly distributed in the specified range.

2) Choose the constant value of $CO_2$ for this period. If the vehicle type is Diesel, the probability follows truncated normal distribution between 1 and 14% with mean of 7% and variance of 6%. If the vehicle type is Gasoline, the probability follows truncated normal distribution between 3 and 14% with mean of 13% and variance of 2%. These parameters reflect the fact that the $CO_2$ concentration in gasoline exhaust is close to constant because the engines typically operate

90  at constant air-to-fuel ratio, whereas in the diesel engines, the air-to-fuel ratio is load dependent.

3) Assign the concentration value for this period by generating a smooth transition between the previous value and the new value with Eq. S9.

4) Start a new period following this period and define the next period by moving back to step 1. Repeat the steps until the end of the cycle is reached.

95  5) For all engine off periods that were defined when assigning the engine exhaust flow rate, assign $CO_2$ concentration of 0%. This simulates a sampling system where the OFR is always sampling zero air when the engine is off.

Hydrocarbons:

1) Choose whether the period of constant value for HC concentration is a stable period (duration between 25 s and 100 s) or a short period (duration between 2 s and 25 s). The probability for a stable period is 0.1 and the probability for duration

100  is uniformly distributed in the specified range.

2) Choose the constant value of HC for this period. The HC value is either low (0-10 ppm), medium (15-200 ppm) or high (500-4000 ppm), reflecting the observed concentrations in cold- and hot-start NEDC for the measured gasoline vehicle. The probability is 0.513 for low concentration, 0.48 for medium concentration and 0.007 for high concentration. In case of low concentration, the probability follows truncated normal distribution between 0 and 10 ppm with mean of 1 ppm

105  and variance of 10 ppm. In case of medium concentration, the probability follows truncated normal distribution between 15 ppm and 200 ppm with mean of 30 ppm and variance of 60 ppm. In case of high concentration, the probability follows truncated normal distribution between 500 and 4000 ppm with mean of 1000 ppm and variance of 1000 ppm.

3) Assign the concentration value for this period by generating a smooth transition between the previous value and the new value with Eq. S9.

110    4) Start a new period following this period and define the next period by moving back to step 1. Repeat the steps until the end of the cycle is reached.

5) For all engine off periods that were defined when assigning the engine exhaust flow rate, assign HC concentration of 0 ppm. This simulates a sampling system where the OFR is always sampling zero air when the engine is off.

A smooth transition between two different values is generated with the following equation:

$$[C](t) = \frac{[C]_0 e^{3 \cdot 1.5} + [C]_f e^{1.5 \cdot t}}{e^{3 \cdot 1.5} + e^{1.5 \cdot t}}, \tag{S9}$$

where $[C]_0$ is the previous value and $[C]_f$ is the new value.

Measured SOA (downstream of the internal ejector diluter, dilution ratio 8.5) with 'low UV' setting compared to modeled SOA. The modeled SOA is calculated by: $[SOA] = [HC] \cdot Y$, where HC is a convolved 10 square pulse of toluene divided with the dilution ratio, and $Y$ is determined from steady input experiments. Toluene concentration upstream DOFR during the pulse was 398 ppb, measured from a steady input experiment. The square pulse was convolved with transfer function. Average flow rate was 6.0 slpm.

**Table S1.** Parameters for calculating OFR transfer functions.

| Reactor | DOFR | | | | | | PAM |
|---|---|---|---|---|---|---|---|
| Gas | $CO_2$ | | | Toluene | | | $CO_2$ |
| UV lamps | off | low | high | off | low | high | on |
| $f_1$ | 0.3301 | 0.5438 | 0.2429 | 0.4799 | 0.5877 | 0.1391 | 0.1357 |
| $f_2$ | 0.6699 | 0.4562 | 0.7571 | 0.5201 | 0.4123 | 0.8609 | 0.3098 |
| $f_3$ | - | - | - | - | - | - | 0.5545 |
| $Pe_1$ | 70.2468 | 59.9304 | 185.0773 | 34.5126 | 40.2907 | 249.9402 | 31.8016 |
| $Pe_2$ | 13.7971 | 13.9073 | 9.3947 | 13.4908 | 24.6792 | 11.7453 | 9.8594 |
| $Pe_3$ | - | - | - | - | - | - | 6.5239 |
| $\tau_1$ (s) | 24.7862 | 27.1867 | 18.7578 | 31.1628 | 28.3984 | 22.1192 | 33.7762 |
| $\tau_2$ (s) | 37.3938 | 49.9008 | 34.9837 | 47.5170 | 54.6086 | 37.5168 | 59.6120 |
| $\tau_3$ (s) | - | - | - | - | - | - | 159.0658 |

[Figure]

**Figure S2.** The measurement setup (DOFR dimensions not to scale). The blue circles depict the DOFR UV lamps. The flows were controlled with mass flow controllers (MFCs; Alicat Scientific). The ozone was generated with an UV lamp (Model 1000, Jelight Company Inc.) and measured with Model 205 analyzer (2B Technologies).

[Figure]

**Figure S3.** Time series of exhaust flow rate ($Q_{exh}$), tailpipe and OFR outlet concentrations of $CO_2$ (a) and SOA (b) in cold-start NEDC, and the cumulative emissions of $CO_2$ (c) and SOA (d). The OFR data is simulated based on tailpipe concentrations and OFR transfer functions, and the SOA concentration refers to HC concentration multiplied with $Y$. All OFR data is delay corrected.

[Figure]

**Figure S4.** Total SOA PFs of subcycles and full driving cycle (a), and time series of  reference SOA PF and SOA PFs determined from OFR measurements (b) in cold-start NEDC. The integrated SOA PF in panel (a) is calculated by normalizing the SOA emission to true $CO_2$ emission, whereas for the momentary SOA PF in panel (b), the SOA concentration is normalized to OFR $CO_2$ concentration. The PFs in both panels are calculated for semi-synthetic SOA data that is linearly proportional to the measured HC concentration in the tailpipe. CSUDC, HUDC and EUDC represent approximately 400 s subcycles within the full cycle. Note logarithmic axis scale in panel (b).

[Figure]

**Figure S5.** DOFR transfer functions for $CO_2$ and toluene with different UV lamp settings. Mean flow rate was 6.8 slpm for $CO_2$ experiments and 6.0 slpm for toluene experiments. According to Dekati, the the transfer function of the current DOFR model consists of a single peak instead of the double peak observed here with the prototype model when the UV lamps were on.

[Figure]

**Figure S6.** The measured DOFR output for 10 s input pulses of $CO_2$ and the simulated output, which is a 10 s square pulse convolved with the transfer function corresponding to the UV setting.

[Figure]

**Figure S7.** The measured DOFR output for 10 s input pulses of toluene and the simulated output, which is a 10 s square pulse convolved with either the $CO_2$ or toluene transfer function corresponding to the UV setting.

[Figure]

**Figure S8.** Best fit transfer function for PAM (a), and a 10 s square pulse of $CO_2$ convolved with the transfer function (b). The experimental data origins from measurements by Lambe et al. (2011), where the PAM UV lamps were on and the ring flow was used.

[Figure]

**Figure S9.** Deconvolution performance test for 6 repetitions of a 10 s toluene pulse input with 'low UV' setting. DOFR output is the actual SOA mass measured with ELPI downstream the internal ejector diluter (dilution ratio 8.5). Input is the square pulse of toluene multiplied with the SOA yield (determined from steady-state experiments) and divided by the dilution ratio. Deconvolved is the result of deconvolution of DOFR output (using $CO_2$ transfer function). Deconvolution overestimates the peak height and underestimates the duration. This is probably because the $CO_2$ transfer function does not perfectly represent the SOA formation dynamics, as observed in Fig. S1a.

[Figure]

**Figure S10.** The ratio of OFR PF to  reference PF when using standard calculation method with different delay correction constants ($\tau$ $\tau_r$). The PFs are calculated for 10 s bins in the driving cycles and different delay correction constants are normalized to $\tau_{peak}$.

[Figure]

**Figure S11.** The effect of calculation bin duration on OFR PF accuracy for hot-start and cold-start NEDC.

[Figure]

**Figure S12.** The effect of calculation bin duration on OFR PF accuracy for 10000 synthetic driving cycles.

[Figure]

**Figure S13.** Comparison of OFR total PF to the  reference PF, when using CVS method (Eq. 11) and sampling zero air after the cycle ends. The driving cycle here is cold NEDC, and the cycle ends at 0 s. The mean residence times of the OFRs ( $\tau_{mean}$) are shown with dashed lines. The data is not corrected for OFR delay, as this is not necessary for the CVS method when calculating the full cycle PF. In contrast, the delay correction will result in some error in the full cycle PF.

[Figure]

**Figure S14.** NEDC divided into bins representing different driving conditions.

[Figure]

**Figure S15.** Correlations between OFR PFs and  reference PFs using the standard PF calculation method. The data corresponds to the histograms in Fig. 4. Note the logarithmic scale in panels (a)-(d).

[Figure]

**Figure S16.**  Reference SOA concentration ($[HC] \cdot Y$) compared to deconvolved OFR SOA signals. For OFRs, the product $[HC] \cdot Y$ is first convolved with the OFR transfer function and then deconvolved using the same transfer function.

[Figure]

**Figure S17.** Examples of synthetic driving cycles generated with the algorithm described in Sect. S4.